# Ferroptosis, Inflammation, and Microbiome Alterations in the Intestine in the Göttingen Minipig Model of Hematopoietic-Acute Radiation Syndrome

**DOI:** 10.3390/ijms25084535

**Published:** 2024-04-20

**Authors:** Timothy Horseman, W. Bradley Rittase, John E. Slaven, Dmitry T. Bradfield, Andrew M. Frank, Joseph A. Anderson, Evelyn C. Hays, Andrew C. Ott, Anjali E. Thomas, Alison R. Huppmann, Sang-Ho Lee, David M. Burmeister, Regina M. Day

**Affiliations:** 1Department of Medicine, Uniformed Services University of the Health Sciences, Bethesda, MD 20814, USA; timothy.horseman@usuhs.edu (T.H.); david.burmeister@usuhs.edu (D.M.B.); 2Department of Pharmacology and Molecular Therapeutics, Uniformed Services University of the Health Sciences, Bethesda, MD 20814, USA; w-bradley.rittase.ctr@usuhs.edu (W.B.R.); john.slaven.ctr@usuhs.edu (J.E.S.); dmitry.bradfield.ctr@usuhs.edu (D.T.B.);; 3Department of Anatomy, Physiology and Genetics, Uniformed Services University of the Health Sciences, Bethesda, MD 20814, USA; andrew.frank.ctr@usuhs.edu; 4Henry M. Jackson Foundation for the Advancement of Military Medicine, Inc., Bethesda, MD 20817, USA; 5Comparative Pathology Division, Department of Laboratory Animal Resources, Uniformed Services University of the Health Sciences, Bethesda, MD 20814, USA; 6Department of Biomedical Sciences, University of South Carolina School of Medicine, Greenville, SC 29605, USA; huppmann@greenvillemed.sc.edu; 7Pathology Department, Research Services, Naval Medical Research Center, Silver Spring, MD 20910, USA; sang.h.lee52.mil@health.mil

**Keywords:** acute radiation syndrome, H-ARS, gastrointestinal injury, captopril, microbiome, minipigs, inflammation, ferroptosis

## Abstract

Hematopoietic acute radiation syndrome (H-ARS) involves injury to multiple organ systems following total body irradiation (TBI). Our laboratory demonstrated that captopril, an angiotensin-converting enzyme inhibitor, mitigates H-ARS in Göttingen minipigs, with improved survival and hematopoietic recovery, as well as the suppression of acute inflammation. However, the effects of captopril on the gastrointestinal (GI) system after TBI are not well known. We used a Göttingen minipig H-ARS model to investigate captopril’s effects on the GI following TBI (^60^Co 1.79 or 1.80 Gy, 0.42–0.48 Gy/min), with endpoints at 6 or 35 days. The vehicle or captopril (0.96 mg/kg) was administered orally twice daily for 12 days, starting 4 h post-irradiation. Ilea were harvested for histological, protein, and RNA analyses. TBI increased congestion and mucosa erosion and hemorrhage, which were modulated by captopril. GPX-4 and *SLC7A11* were downregulated post-irradiation, consistent with ferroptosis at 6 and 35 days post-irradiation in all groups. Interestingly, p21/waf1 increased at 6 days in vehicle-treated but not captopril-treated animals. An RT-qPCR analysis showed that radiation increased the gene expression of inflammatory cytokines *IL1B*, *TNFA*, *CCL2, IL18*, and *CXCL8*, and the inflammasome component *NLRP3*. Captopril suppressed radiation-induced *IL1B* and *TNFA*. Rectal microbiome analysis showed that 1 day of captopril treatment with radiation decreased overall diversity, with increased Proteobacteria phyla and Escherichia genera. By 6 days, captopril increased the relative abundance of Enterococcus, previously associated with improved H-ARS survival in mice. Our data suggest that captopril mitigates senescence, some inflammation, and microbiome alterations, but not ferroptosis markers in the intestine following TBI.

## 1. Introduction

Acute radiation syndrome (ARS) resulting in multi-organ injury is a potentially fatal effect of total body irradiation (TBI) [1,2]. The specific impact of radiation depends upon the total radiation dose, the dose rate of the exposure, and the areas of the body exposed [2,3]. Extensive research on the natural history of ARS has shown that there are several distinct subsyndromes (e.g., hematopoietic ARS [H-ARS] and gastrointestinal ARS [GI-ARS]), each with defined dose- and time-dependent characteristics [1]. The total radiation dose exposure required to induce the individual subsyndromes varies with the radiation sensitivity of the organ systems [1], with the hematopoietic system displaying the highest sensitivity to radiation. H-ARS is associated with a loss of mature blood cells in the circulation and a loss of hematopoietic stem and progenitor cells in the bone marrow [1]. The primary cause of mortality, which occurs ~30–35 days post-irradiation, is due to the sensitivity of the hematopoietic system to radiation in all animal models [4,5]. Beyond hematopoietic insufficiency, H-ARS is characterized by infection from immune suppression, acute systemic inflammation, and coagulation dysfunction [1,6,7,8,9,10,11,12].

Subacute radiation damage to the GI system is recognized to contribute to mortality from H-ARS [13,14,15,16]. In the Göttingen minipig model, subacute GI injury was observed at doses of radiation that induced H-ARS (1.7–1.9 Gy, 0.6 Gy/min), with symptoms of GI injury (dry stools, anorexia, and no weight gain) observed in most animals over the time course from 2 h to 14 days post-irradiation, but without diarrhea or GI ulceration [17]. More severe GI injury was observed from 3 to 6 days following 2.2–4 Gy (0.6 Gy/min) TBI, a dose of radiation associated with H-ARS but not GI-ARS [18]; GI effects included villi blunting and fusion, intestinal hemorrhages, and decreased serum citrulline (indicator of the loss of GI barrier integrity) [17,18]. Signs of acute sickness following H-ARS irradiation (weakness, facial edema, petechiae, and desquamation in areas of the skin) and initial peripheral blood cell recovery were observed by 35 days post-irradiation [17]. The resolution of subacute GI injuries at late time points was not completely defined. GI damage from TBI is believed to contribute to H-ARS mortality through the loss of nutritional absorption capacity, as well as the loss of integrity of the intestinal mucosal barrier, allowing bacterial translocation, infection, an exaggerated inflammatory response, and sepsis [13,14,15,16]. Murine models of H-ARS and GI-ARS showed that TBI, as well as high-dose abdominal irradiation, caused decreased crypt depth (decreased stem cell survival), alterations in villi morphology (villus blunting and fusion), and intestinal wall thickening [14,15]. The inflammatory response of the GI tract following radiation exposure includes the local production of pro-inflammatory cytokines by transcription factor activation, as well as through the upregulation/activation of inflammasomes, which can proteolytically cleave and secrete pro-inflammatory cytokines [14,15,19,20]. Recent studies have also implicated iron-regulated oxidative stress or, independently, ferroptosis (a type of programmed cell death) as factors following TBI [21,22].

Preclinical studies suggested that the enhanced survival from H-ARS with the use of antibiotics is attributed to the mitigation of infections [14,23]. Importantly, not all gut bacteria are associated with increased mortality following TBI. Some gut microbiome studies suggest that specific bacteria families (e.g., *Lachnospiraceae*, *Enterococcaceae*, and *Lactobacillaceae*) may contribute to improved survival from H-ARS [16,24,25]. The downstream metabolites produced by these bacterial families (especially propionate, valeric acid, and tryptophan metabolites) were demonstrated to have radioprotective effects [16,26]. However, the large majority of microbiome studies have been performed in rodent models that have a different microbiome composition compared to humans or large animals [27,28,29,30].

The renin–angiotensin–aldosterone system (RAAS) has been shown to be an effective target for the prevention and mitigation of H-ARS, especially angiotensin II (Ang II) and angiotensin-converting enzyme (ACE) inhibitors [31,32,33,34,35,36]. The RAAS is a critical regulator of blood pressure and blood volume, and it has recently been shown to play a role in hematopoietic progenitor proliferation and differentiation [36,37,38,39,40]. Our laboratory demonstrated that the orally available ACE inhibitor captopril can reduce the cycling of hematopoietic progenitors in vivo [32,33], and we demonstrated that captopril is an effective countermeasure in murine and Göttingen minipig models of H-ARS [32,33,34,40]. We showed that captopril-reduced mortality following TBI was associated with improved mature blood cell recovery, improved bone marrow and hematopoietic progenitor recovery, suppression of the acute inflammatory response, and the reduction of radiation coagulopathy [32,33,34,35]. Here we characterize the effects of TBI and the effects of captopril on the ileum in a Göttingen minipig model of H-ARS, including structural alterations, alterations in genes and proteins associated with iron binding and ferroptosis, and inflammatory gene expression. We selected an early time point of 6 days, previously shown to correspond with early radiation injuries in the GI; and a late time point of 35 days was selected at which the hematopoietic system begins to display recovery. We investigated early changes in the microbiome from 1 to 6 days post-irradiation, as the gut microbiota can be associated with inflammatory responses, as well as ferroptosis in the intestine42,43.

## 2. Results

### 2.1. Histological Analysis of Captopril Effects on Radiation Sequelae in the Intestine

Previous studies showed that TBI induced hemorrhages and/or petechiation in a number of tissues, including skin, lung, heart, kidney, liver, intestines, and stomach ~14–35 days post-irradiation in Göttingen minipigs [17]. We found that, in irradiated minipigs administered the vehicle, 50% of animals displayed areas of ecchymoses on the surface of the large intestine, small intestine, and peritoneum by 35 days post-irradiation (8/16 animals) (Figure 1). Captopril treatment reduced the incidence of ecchymoses on internal tissues to 25% of the animals (3/12 animals).

An analysis of H&E- and Alcian blue-stained histological sections showed that normal GI structures were present in all groups after radiation, suggesting that the impact on the GI tract was likely subacute (Figure 2A). At 6 days in radiation + vehicle animals, there was a none to slight increase in mitosis, mononuclear infiltration, and congestion, and there was no detectable mucosal erosion or hemorrhage (Figure 2B–F). At 35 days in radiation + vehicle animals, there was a none to slight increase in mitosis, mononuclear infiltration, and congestion. There was a slight to mild mucosal hemorrhage, and there was mild mucosal erosion. In the radiation + captopril group at 6 days, there was a none to slight mononuclear infiltration, congestion, and mucosal erosion, and there was slight to mild increased mitosis. There was no detectable hemorrhage at this time point. In the radiation + captopril group at 35 days, there was slight congestion, but there was no evidence of increased mitosis, mononuclear infiltration, mucosal erosion, or mucosal hemorrhage. Thus, overall, there appeared to be a trend toward reduced histological abnormalities in the ileal tissues from the radiation + captopril group at 35 days. In all groups, there was no evidence of neutrophilic infiltration in the lamina propria, loss of crypts, submucosal hemorrhage, or lymphoplasmacytic or neutrophilic submucosal infiltration. In some images, shortened villi with increased width were observed, but this did not reach significance (Figure 2G,H). Radiation exposure slightly reduced crypt depths (~25–30%) in both vehicle- and captopril-treated animals, but this also did not reach statistical significance (Figure 2I). TBI had a significant effect on intestinal wall thickness, with a ~50% increased thickness in radiation + vehicle animals at both time points combined (*p* = 0.023) but not in the radiation + captopril time points (Figure 2J). It appeared that, following radiation, there was a reduced number of goblet cells at the tips of the villi; these cells produce a protective mucus layer with multiple functions, including the prevention of bacterial translocation. However, analysis showed that the reduction of goblet cells at the top 1/3 of the villi was not significant in radiation + vehicle groups (Figure 2K).

### 2.2. Iron-Binding Proteins and Ferroptosis-Related Enzymes Are Altered after Radiation in the GI but Are Not Significantly Affected by Captopril

We investigated the regulation of iron-binding proteins and enzymes that regulate ferroptosis in Göttingen minipig intestines [41,42,43,44,45,46,47]. Ferritin is the main cytosolic iron-storage protein in many cell types There was a trend toward increased ferritin in radiation + vehicle and radiation + captopril animals at day 6, but there was no significance in the post hoc analysis (Figure 3A). The *SLC40A1* gene encoding the iron-exporting protein ferroportin showed a non-significant trend toward downregulation (77–83%) in the radiation + vehicle and radiation + captopril groups at 6 and 35 days (Figure 3B). These data suggest that there are trends toward increased iron storage and decreased iron export post-irradiation. Note that there was no significant effect of captopril on the levels of ferritin or *SLC40A1* in sham-irradiated animals.

Lipid peroxidation has been shown to occur in mouse intestinal tissues following ionizing radiation exposure [21]. The levels of lipid peroxidation were increased ~20–30% over the control levels and were detectable at 3 days, but not 6 days, post-irradiation [21]. We investigated levels of MDA-linked protein levels as an indirect measure of lipid peroxidation [42,43]. In sham-irradiated intestinal tissue, the basal levels of MDA-linked proteins were relatively low compared with the liver and kidney, but were similar to levels of MDA-linked proteins in the lung (Figure 4A). At 6 days post-irradiation, we observed some increase in MDA-linked proteins of 25 and 40 kDa MW, but the levels of these proteins were not consistent between samples. We therefore investigated other markers of ferroptosis. Intestinal glutathione peroxidase-4 (GPX-4; gene *GPX4*), the sole mammalian enzyme that reduces phospholipid oxides to alcohols, and solute carrier family 7 member 11, the cystine transporter (gene *SLC7A11*), are inhibitors of ferroptosis; the inhibition of either of these two enzymes is sufficient to induce ferroptosis [42,46,47]. GPX-4 protein levels were significantly reduced by ~70–83% compared to the sham + vehicle in all groups (6 days: radiation + vehicle *p* = 0.015, radiation + captopril *p* = 0.026; 35 days: radiation + vehicle *p* = 0.007, radiation + captopril *p* = 0.012) (Figure 4A). Likewise, *GPX4* gene expression was significantly decreased at 6 days (~90–94%) in both the radiation + vehicle (*p* = 0.013) and radiation + captopril (*p* = 0.003) groups compared to the sham + vehicle (Figure 4B). *GPX4* showed a trend toward reduced expression in both groups at day 35. We also observed a ~60–75% downregulation of *SLC7A11* at day 6 post-irradiation in both radiation + vehicle (*p* = 0.006) and radiation + captropril (*p* = 0.001) animals compared to sham + vehicle (Figure 4C). *SLC7A11* showed a trend in reduction through day 35 post-irradiation in the radiation + captopril group. These data indicate a reduction in both GPX-4 protein and gene expression and *SLC7A11* gene expression in vehicle and captopril groups, which would favor ferroptosis. Captopril did not significantly change GPX-4 protein levels or *GPX4* gene expression in sham-irradiated animals. Captopril treatment did not induce a significant increase in the level of *SLC7A11*.

To further examine the expression of genes involved in ferroptosis, we examined Nrf2 (*NRF2*), a transcription factor which promotes the expression of antioxidant response enzymes and the antioxidant enzyme heme oxygenase-1 (*HMOX1*). Both of these negatively regulate ferroptosis [44,46,47]. At 6 days, there was a trend toward reduced *NRF2* in radiation + vehicle animals, and there was a significant (77%) reduction in *NRF2* in radiation + captopril animals (*p* = 0.026) (Figure 4D). *NRF2* showed a trend toward reduced levels in both groups at 35 days post-irradiation. *HMOX1* expression was significantly downregulated 95–97% at 6 days post-irradiation in both groups (vehicle, *p* = 0.0073; captopril, *p* = 0.0017) compared to the sham + vehicle (Figure 4E). There was also a significant reduction of *HMOX1* (94%) at 35 days in the radiation + captopril group (*p* = 0.026), and a trend toward reduced levels at 35 days in the radiation + vehicle group. Captopril treatment had no effect on *NRF2* or *HMOX1* expression in sham-irradiated animals.

Ferroptosis can occur in the absence or presence of activated caspases, including caspase-3, depending upon cross-talk with other cell death pathways [42,45,46,47]. Interestingly, the intestine was shown to have constitutively high levels of active caspase-3 under homeostatic conditions [48], although the biological function of constitutively activated caspase-3 is not understood. In contrast with studies showing the activation of caspase-3 following TBI, we found that caspase-3 activity was 50% lower at 6 days in radiation + vehicle (*p* = 0.049) and radiation + captopril (*p* = 0.049) (Appendix A). Cox-2 (encoded by *PTGS2*) was demonstrated in some tissues to serve as a marker for ferroptosis, although Cox-2 does not appear to have a functional role in this process [41,45]. *PTGS2* is also constitutively expressed in the intestinal wall and may play a role in tissue homeostasis [49]. As observed for caspase-3 activity, there was an 83–89% decrease in *PTGS2* expression at 6 days in both groups (*p* = 0.009 and *p* = 0.004 in radiation + vehicle and radiation + captopril, respectively) and at 35 days in the radiation + captopril group (87% reduction, *p* = 0.039) compared to the sham + vehicle (Appendix A). Captopril had no significant effects on caspase-3 activation or *PTGS2* expression in sham-irradiated animals.

### 2.3. Captopril Mitigates Radiation-Induced Upregulation of p21/waf1 and Some Pro-Inflammatory Cytokine Expression in the GI

We investigated p21/waf1 upregulation, a marker of cell cycle arrest, at 6 and 35 days post-irradiation (Figure 5A). Radiation increased p21/waf1 protein levels ~4-fold at 6 days in radiation + vehicle animals (*p* = 0.0014) but not in radiation + captopril (*p* = 0.73) animals, compared to the sham + vehicle. Moreover, captopril inhibited the radiation-induced upregulation of p21/waf1 at 6 days (*p* < 0.0001). p21/waf1 returned to sham + vehicle levels by 35 days in both groups. There was no significant difference in p21/waf1 levels between the sham + vehicle and sham + captopril groups.

There was a general increase in local pro-inflammatory cytokine transcription in the intestinal tissues after TBI, which was significant only at the 6 day time point (Figure 5B–H). *IL1B* showed a trend toward increased expression (~2.7-fold), and *TNFA* expression was increased ~3-fold in the radiation + vehicle group at 6 days (*p* = 0.046). At the 6-day time point, animals treated with captopril did not display an increased expression of *IL1B* or *TNFA* compared to the sham + vehicle animals. Also at 6 days, chemokine (C-C motif) ligand 2 (*CCL2*) was elevated (~4–7-fold) in both the radiation + vehicle (*p* = 0.024) and radiation + captopril (*p* = 0.029) groups, compared to the sham + vehicle. There was a trend toward increased *IL6* expression after TBI, especially in the radiation + vehicle group at 6 days, but this did not reach statistical significance. Radiation increased the expression of *IL18* ~4-fold in radiation + captopril animals (*p* = 0.01) but showed only a trend toward increased expression in radiation + vehicle animals (*p* = 0.055) at day 6, compared with sham + vehicle ones. Similarly, TBI increased the expression of *CXCL8* ~3-fold at day 6 in radiation + vehicle animals (*p* = 0.011) and in radiation + captopril animals (*p* = 0.0034), compared to sham + vehicle ones. The expression of *NLRP3*, a critical component of the inflammasome that is required for the processing and secretion of some cytokines, was also increased (~4–5-fold) at 6 days in the radiation + vehicle (*p* = 0.0003) and radiation + captopril (*p* = 0.0001) groups, compared with the sham + vehicle group. Note that there was no significant difference in *IL1B*, *TNFA*, *CCL2*, *IL6*, *IL18*, or *CXCL8* in the sham + captopril compared with the sham + vehicle group, and there no was significant level of increase in *NLRP3* expression by captopril in the sham-irradiated animals.

Lastly, we investigated anti-inflammatory *IL10* expression (Figure 5I). At day 6, *IL10* was increased in both radiation + vehicle animals (*p* = 0.014) and radiation + captopril animals (*p* = 0.018), compared with sham + vehicle ones. While there was no significant effect of captopril on *IL10* expression compared to the radiation + vehicle groups, there was a significant effect associated with time, with both vehicle- (*p* = 0.014) and captopril- (*p* = 0.014) treated animals showing a higher expression at day 6 compared to day 35 (Figure 5I). Interestingly, the sham + captopril expression of *IL10* was increased ~2-fold compared with sham + vehicle animals (*p* = 0.022).

### 2.4. Captopril Effects on Radiation-Induced Alterations of the Gut Microbiome

We aimed to elucidate acute alterations in the gut microbiota following captopril administration post-irradiation, compared to radiation exposure without treatment, using shallow shotgun sequencing. The alpha diversity of bacteria within the samples was measured using Shannon diversity and Faith’s phylogenetic diversity (Faith PD) (Figure 6A). At day 1 post-irradiation, the vehicle treatment group had a significantly higher Shannon diversity than captopril-treated animals (*p* < 0.05). The Shannon diversity decrease was not significantly different from baseline in both cohorts at 3 and 6 days post-irradiation. The Faith PD for each group did not show significant differences across time points. 

To evaluate the longitudinal microbial community composition between cohorts, we evaluated four beta diversity metrics. The Bray–Curtis and Jaccard are both non-phylogenetic beta diversity measures, while the weighted and unweighted UniFrac are both phylogenetic metrics with the weighted UniFrac and Jaccard metrics considering microbial abundance (Figure 6B). The vehicle-treated, irradiated group clustered significantly differently than the captopril-treated, irradiated animals with each beta diversity measure (*p* < 0.05). At day 1 post-irradiation, both treatment groups had significant community shifts with the Bray–Curtis (*p* = 0.025) and weighted UniFrac (*p* = 0.033) metrics (Figure 5B). Axis 1 illustrates a majority of the variation in each beta diversity metric, with captopril-treated samples shifting further down this axis.

The relative and differential abundance of rectal flora was evaluated at the phylum and genus bacterial taxonomy levels (Figure 6C, Table 1 and Table 2). Overall, Firmicutes, Bacteriodetes, Proteobacteria, and Actinobacteria were not significantly altered in the radiation + vehicle group at any time point. However, there was a ~10-fold increase in Verrucomicrobia in the radiation + vehicle group on day 1 post-radiation (*p* = 0.022) compared to both baselines together; this increase was significantly reduced in the radiation + captopril group at day 1 (*p* = 0.016). The radiation + captopril group was also associated with trends toward increased Proteobacteria (change from baseline, 270%) and decreased Firmicutes (from baseline, 43%) at day 1. An evaluation of genus level taxonomy revealed that radiation exposure led to a non-significant trends toward decreased beneficial normal microbiota such as Lactobacillus (~5–10-fold reduction by day 6) and Faecalibacterium (~3–10-fold reduction by day 6). In both groups, these trends coincided with non-significant trends toward the enrichment of genera associated with increased inflammation, Bacteroides (~2-fold increase by day 6), and Escherichia (~7–15-fold increase by day 6). Captopril administration post-irradiation did not appear to significantly alter the relative abundance of most gut microbiota.

At the phyla level, the swine gut microbiota is dominated by Bacteroidetes and Firmicutes similar to the human gut. The Firmicutes/Bacteroidetes (F/B) ratio has historically been touted as a measure of gut microbiome stability and health. In general, irradiation decreased the F/B ratio; however, captopril-treated animals had an F/B non-significantly closer to the baseline than the vehicle-treated group on day 6 following radiation (vehicle-treated, baseline to day 6, *p* = 0.52; captopril-treated, baseline to day 6, *p* > 0.99) (Appendix A).

## 3. Discussion

Exposure to total body irradiation can lead to multi-organ injury, and it has been recognized that damage to the GI contributes significantly to the morbidity and mortality in H-ARS following TBI [14,15,16,17]. Multiple effects of radiation on the intestines have been recognized, including changes in tissue structures [15,16,22], oxidative stress, iron-dependent programmed cell death (ferroptosis) [22], increased local production of cytokines [21,22,43,50,51], and alterations in the gut microbiota [52]. Swine are increasingly being used for medical research, a choice which is supported by anatomical and immunological similarities between swine and humans [53]. In this study, we characterized the effects of TBI on GI structure, ferroptosis, and microbiome composition in a Göttingen minipig model of H-ARS.

Analysis of the intestine histopathology suggested that the GI injury was subacute. The loss of villi, decreased length of villi, and increased width of villi were previously reported as indicators of acute GI radiation injury [15,16,22]. A reduction in the villi surface area can lead to reduced nutrient absorption. Additionally, damaged membranes with a reduced production of mucus (due to reduced goblet cells) can increase the susceptibility to microbial translocation, which may contribute to reduced survival post-irradiation [16]. The increased thickness of the intestinal wall at short time points may be an indication of local inflammation, while an increased wall thickness at later time points could be indicative of fibrotic remodeling [16]. The subacute GI effects of 1.79–1.80 Gy exposure that we found in minipigs are similar to findings in a previous study of mice, where a 2 Gy TBI resulted in only a 20% decrease in villi height and a ~15% reduction in the crypt depth [21].

A number of recent studies have suggested that ferroptosis is a secondary toxicity following TBI [21,43,54,55,56]. Our laboratory and others demonstrated that, following TBI, iron is deposited in a number of tissues, following the release of iron from the hemolysis of red blood cells and reticulocytes, and that ferroptosis can be activated [43,54,55,56]. Ferroptosis was demonstrated in the intestines of mice following a 2 Gy TBI [22], which was associated with an increased expression of arachidonate 15-lipoxygenase (ALOX15), an enzyme believed to be involved in polyunsaturated fatty acid-phospholipid peroxidation [57]. In our Göttingen minipig study, TBI induced trends toward an increase in the levels of the iron-storage protein ferritin and a decrease in *SLC40A1*, the mRNA of the iron-export protein ferroportin. Coinciding with the trends of increased iron storage and reduced iron export, GPX-4 protein and *GPX4* mRNA levels decreased. Studies showed that the genetic or pharmacological inactivation of either GPX-4 or SLC7A11 is sufficient to induce ferroptosis [45,46]. Therefore, both of these proteins being decreased would be consistent with conditions favoring ferroptosis [45,46]. Assays for lipid oxidation and mitochondrial morphological changes, with additional time points of tissues for analysis, would strengthen the findings of radiation-induced ferroptosis in the minipig intestine following TBI.

Local and systemic inflammation are hallmarks of radiation exposure, even at sublethal exposures [1,9,32,34,35,58]. The generation of acute inflammatory responses that we observed following total body irradiation exposure is likely due to a combination of local cellular injury/death and the loss of hematopoietic cells, triggering the upregulation of cytokines to induce the cycling/development of hematopoietic stem and progenitor cells [32,34,35,59,60]. A sustained upregulation of p21/waf1 and pro-inflammatory cytokines can occur through radiation-induced accelerated senescence, causing the activation of the senescence-associated secretory phenotype [61,62]. In contrast, the transient upregulation of p21/waf1, other cell cycle inhibitors, and p53 can occur downstream of DNA damage signaling [62]. Our limited time course study cannot distinguish whether the upregulation of p21/waf1 is sustained or transient, and further studies are needed to conclusively determine whether senescence occurs in the intestine following a 1.79–1.80 Gy TBI in minipigs. However, captopril’s reduction of p21/waf1 suggests either the suppression of DNA damage signaling or the suppression of accelerated senescence in the minipig GI, consistent with our previous findings [35,56,62]. Here, we found that TBI induced the upregulation of the mRNA encoding a wide variety of cytokines in the intestines of minipigs, consistent with the activation of local inflammation and intestinal inflammasome activation. The NLRP3 inflammasome has been shown to be implicated in GI-ARS previously, and has also been shown to be affected by the gut microbiota [20,63]. Captopril reduced the tissue expression of some but not all of these cytokines, but did not affect the regulation of *NLRP3* or *IL18*. Future studies with more time points and increased animal numbers could provide more information regarding accelerated senescence and other types of pathogenic processes.

Radiation exposure has been shown to alter gut microbiota [20,64]. A recent study of gut microbiota in mice found that the Firmicutes/Bacteroidetes ratio decreased and Lactobacillus abundance decreased post-irradiation [65]. The few large animal studies conducted have discordant taxonomic findings [66,67,68]. One study showed opposite trends post-irradiation at the phyla level in minipigs and non-human primates (NHPs), wherein minipigs displayed increased Firmicutes and decreased Bacteroidetes and Proteobacteria [67,68]. At the genus level, most post-irradiation data include decreases in Lactobacillus and Bifidobacterium, with increases in Enterobacteriaceae and Prevotella, but reports of Bacteroides abundance have differed greatly in various studies [66,67,68]. The GI microbiome may also impact survival post-irradiation, although evidence for this was obtained only from rodent models. For example, Guo et al. (2020) evaluated the gut microbiota in a mouse model of lethal irradiation, identifying a class of survivors with a distinct microbial profile that included increased Lachnospiraceae and Enterococcaceae [16]. In our study, we observed an increase in Bacteroidetes in the vehicle-treated group post-irradiation, but a decrease with captopril treatment. Our study also showed that radiation decreased Firmicutes and increased Proteobacteria, independent of captopril treatment.

The microbiome has been demonstrated to affect both inflammation and ferroptosis in the GI [69,70,71,72,73,74]. Increased levels of *E. coli* were demonstrated in murine studies to suppress GI ferroportin expression, leading to increased intracellular iron and increased reactive iron [69]. Additionally, studies of inflammatory bowel disease have shown an association between increased *E. coli* levels in the intestinal microbiome and increased inflammation [71]. Our data show a trend toward increased levels of Escherichia post-irradiation (up to 7–10-fold at 6 days post-irradiation), which could potentially contribute to ferroptosis and inflammation. A previous study of gut microbiome diversity in normal Wistar Kyoto rats showed that captopril treatment did not significantly alter alpha diversity but did alter the bacterial composition after a period of four weeks of treatment [75]. Similar studies have not yet been performed in minipigs.

Our swine model represents a novel longitudinal evaluation of the gut microbiome following TBI and captopril treatment, using unbiased non-amplicon-based whole genome sequencing. Our data suggest that captopril does not significantly mitigate the effects of radiation on the microbiome. At 6 days post-irradiation, we observed that captopril treatment resulted in an increased relative abundance of Enterococcus, a genus previously associated with improved survival from ARS in mice [16,25,26]. We believe these results warrant further study, to include higher radiation doses, as well as the investigation of microbiome-related metabolites [16]. There has been recent evidence of a link between the gut microbiota and cellular ferroptosis with intestinal bacteria thought to regulate iron overload and ferroptosis directly or through their metabolites [69]. Furthermore, microbial dysbiosis in the gut has been demonstrated to induce ferroptosis in zebrafish and mice [72,73]. Additionally, Bacteroides have been associated with ferroptosis-related markers [74,76]. Significant GI symptoms would be expected to appear in the subsequent week(s), and an extended time course and varying doses is a limitation of the current study. Additionally, the lack of a captopril-only treatment group and the small sample size (n = 4/group/time point) may result in an underestimate of some of the changes presented due to limited power.

Radiation countermeasure development for the mitigation of H-ARS and GI-ARS is a critical research goal in radiation biology. Because of the ethical issues for the testing and development of radiation countermeasures, the US Food and Drug Administration Animal Rule requires a demonstration of countermeasure efficacy in two animal models, one of which must be a non-rodent [77]. Recent studies suggest that, because of anatomical and physiological similarities, swine recapitulate many of the characteristics of H-ARS in humans [78]. Our laboratory has repeatedly demonstrated the efficacy of captopril as a mitigator of H-ARS in mice and Göttingen minipigs [34,35,40]. Here we demonstrate that captopril mitigates some local inflammation in the GI following TBI. However, additional agents may be required to completely mitigate the effects of radiation on the GI.

## 4. Materials and Methods

### 4.1. Reagents and Chemicals

Reagents were obtained from Sigma-Aldrich (St. Louis, MO, USA) except where indicated.

### 4.2. Animals, Irradiation, and Drug Administration

All animal research was conducted in compliance with guidelines from the National Research Council for the ethical handling of laboratory animals and approved by the Uniformed Services University (USU) and Armed Forces Radiobiology Research Institute (AFRRI) Institutional Animal Care and Use Committees (IACUCs). Housing was as previously described [55]. Male Göttingen swine (4–6 months of age, 8–11 kg upon shipment) were obtained from Marshall BioResources of Marshall Farms Group, Ltd. (North Rose, NY, USA). Animals were group-housed for communal enrichment in a 20 ± 2 °C, 50% ± 10% humidity, and 12:12 light/dark cycle in a facility accredited by the Association for Assessment and Accreditation of Laboratory Animal Care-International (AAALACI). A commercial minipig diet (Harlan Teklad Minipig diet 8753, Madison, WI, USA) was provided twice daily, with water available ad libitum. The swine were housed within the animal facility at USU for 3–6 days to acclimate prior to the study. The swine were randomized into groups (total animal numbers were achieved over the course of 4 separate experiments): (1) sham + vehicle (anesthesia only); (2) radiation + vehicle; (3) radiation + captopril. While under anesthesia (telazol/xylazine, 4.4–2 mg/kg), the swine were positioned in a Panepinto sling and exposed bilaterally to a target total-body dose of 1.79–1.80 Gy ^60^Co irradiation delivered at a dose rate of 0.485–0.502 Gy/min, as previously described [18,35,79,80]. A dose of 1.79 Gy was used for three studies with 35-day endpoints and 1.80 Gy was used for one study with a 6-day endpoint. Both doses of radiation induce hematopoietic ARS but not GI-ARS [17,18,35]. After irradiation, the animals were returned to their housing and allowed to recover from anesthesia while under observation. Captopril (USP grade, Sigma-Aldrich) was dissolved in sterile water and administered orally, 0.96 mg/kg, mixed in ~1 mL of yogurt immediately before administration twice daily. Captopril or the vehicle was administered twice daily during the study, from 4 hours through to 12 days post-irradiation. Animals were weighed three times weekly, and temperatures were obtained under inhaled isoflurane as previously described [35]. Animals exhibiting morbidities were euthanized by intracardiac injection of Euthasol^®^, 4.5 mL/kg (Virbac, Fort Worth, TX, USA), according to current American Veterinary Medical Association guidelines, by a staff veterinarian [35]. Radiation + vehicle and radiation + captopril animals were euthanized either at 6 or 35 days post-irradiation. Sham + vehicle animals were euthanized at 35 days after sham irradiation.

### 4.3. Tissue Collection and Histology

Immediately following euthanasia, a necropsy was performed on each animal to assess the gross pathology and collect tissues. Histological samples from the small intestine (ileum) were obtained roughly 10 inches into the proximal end of the ileum, processed, and stained with hematoxylin and eosin (H&E) and Alcian blue (to detect polysaccharide-containing structures) by Histoserve, Inc. (Germantown, MD, USA). Histological sections were evaluated by pathologists blinded to the treatment groups. Histological images were obtained using a Nikon Eclipse (Melville, NY, USA) with Nikon Elements software (v 4.0). H&E slides were scanned using a Zeiss Axioscan Z1 slide scanner (Carl Zeiss Microscopy, Jena, Saxe-Weimar-Eisenach, Germany), and measurements of villus width, villus length, crypt depth, and intestinal wall thickness, and the quantification of goblet cells, were performed using the associated ZEN 2012 (Blue) software by a trained researcher blinded to the treatment groups (Appendix A). The histological results were scored by a board-certified veterinary pathologist for an increased mitosis of intestinal epithelial cells, mononuclear infiltration of the lamina propria, congestion (characterized by the accumulation of erythrocytes within the blood vessels, causing mild dilation of the vessels), mucosal erosion (characterized by the necrosis or loss of superficial luminal epithelial cells with partial mucosal penetration, but with intact muscularis mucosa), and mucosal hemorrhage. A semi-quantitative scoring system was used: 0 = within normal limits; 1 = slight, barely exceeding normal limits; 2 = mild, lesion easily identified but of limited severity; 3 = moderate, lesion is prominent, but there is significant potential for increased severity; 4 = marked, lesion occupies the majority of the examined tissue with little potential for increased severity.

### 4.4. Microbiome

Rectal swabs were obtained at baseline (prior to irradiation) while animals were under inhaled anesthesia and at days 1, 3, and 6 post-TBI. DNA isolation from the swabs was performed with the FecalPower Pro Kit (Qiagen, Germantown, MD, USA), with the concentration measured for each sample using a Qubit^®^ 4.0 Fluorometer (Life Technologies, Grand Island, NY, USA). Libraries were constructed from the extracted DNA with an adequate yield, using the Illumina DNA PCR-Free workflow without modifications, followed by quantification using the KAPA Library Quantification Kit (KAPA Biosystems, Wilmington, MA, USA) for normalization and pooling. Shallow shotgun sequencing was performed on the Illumina NovaSeq 6000 platform (Illumina, San Diego, CA, USA). Illumina paired-end reads were processed using the Nextflow workflow nf-core/mag (https://github.com/nf-core/mag; v2.1.1 URL (accessed on 9 September 2022) [81]. Quality trimming and adapter removal were performed using fastp (v0.23.3) with default parameters and FASTQC (v0.11.9) to visualize quality metrics for each sample [82]. PhiX and host reads were removed by Bowtie2 (v2.4.5), with mapping to the susScr3 porcine reference genome [83]. Taxonomic classification was performed on the remaining reads with Kraken2 (v2.0.8), using the GTDB_release207_kraken2 database pre-compiled by the authors of Struo2 (v2.3.0) [84,85]. The Kraken2 output was converted from the percent abundance of taxa to relative abundance using Bracken (v2.5.0) [86]. A Biom table from Bracken files was created using Kraken-Biom (v1.0.1) [87]. The Biom table was imported into Quantitative Insights into Microbial Ecology (QIIME2, version 2021.8) for the generation of alpha and beta diversity metrics [88]. Rarefaction for the diversity metric calculation was set to 485,000 sequences per sample. The R (v 4.1.2) package qiime2R (v0.99.6) was used to convert .qza files into a phyloseq object, and the differential abundance was evaluated with DESeq2 (v1.36.0) [89,90].

### 4.5. Western Blotting

Ileum samples obtained post-mortem were snap-frozen and immediately stored at −80 °C. To prepare lysates, approximately 0.2 mL Pierce RIPA buffer (ThermoFisher Scientific, Waltham, MA, USA; #8900) containing a protease and phosphatase inhibitor cocktail, 25 µg each, (Roche Applied Science, Indianapolis, IN, USA) was added to 10 µg of tissue, and samples were homogenized using the Bullet Blender 5E Gold (Next Advance, Troy, NY, USA), using Next Advance Navy Eppendorf Lysis tubes. Subsequently, the tubes were centrifuged at 6000 RCF for 7 min at 4 °C and supernatant retained. The total protein concentration of the supernatant was determined by a BCA Protein Assay (Thermo Fisher Scientific, Waltham, MA, USA). Samples were separated by SDS gel electrophoresis, using 10 or 12% gels poured inhouse. Electrophoresis and electroblotting onto nitrocellulose membranes were performed using the BioRad (BioRad, Hercules, CA, USA). Membranes were blocked with 5% BSA (Santa Cruz Biotechnology, Dallas, TX, USA) in Tris buffered saline (TBS) with 5% Tween (Bio-Rad, Hercules, CA, USA; #17065531XTU; TBST) at room temperature. After blocking, the nitrocellulose membranes were incubated for 16 h at 4 °C with the primary antibody. Primary antibodies used were as follows: anti-full length and cleaved caspase-3 antibody (Cell Signaling, Danvers, MA, USA, #9662, 1:1000); anti-p21/waf1 (Cell Signaling, #2947, 1:1000); β-actin (MilliporeSigma, St. Louis, MO, USA, #A1982, 1:5000); anti-malondialdehyde (MDA) for the detection of modified proteins (Alpha Diagnostic International, San Antonio, TX, USA, #MDA11-S: 1:1000); and anti-GPX-4 (Cell Signaling, #52455, 1:1000). TBST was used to wash blots 3 times, followed by a TBS (VWR Chemicals, Bridgeport, NJ, USA; #97064-338) rinse. Conjugated secondary antibodies (LI-COR, Lincoln, NE, USA; 1:10,000) were used for detection, visualized on the Odyssey system (LI-COR). β-actin was used as a loading control and for normalization.

### 4.6. RNA Isolation and Quantitative PCR Analysis

Intestinal tissues were harvested and immediately placed in RNAlater, following euthanasia. Tissues in RNAlater were stored at −20 °C for 24 h and then at −80 °C for longer periods, according to the manufacturer’s instructions (Qiagen, Germantown, MD, USA). At the time of RNA isolation, tissues were homogenized with an Ultra Turrax homogenizer on ice (Jahnke & Kunkel, Staufen, Germany). RNA was extracted using the Qiagen RNeasy Mini kit with on-column DNase digestion (Qiagen, Valencia, CA, USA) according to manufacturer’s protocol. The quantification and quality determination of isolated RNA was carried out spectroscopically via Nano-drop (ND-1000 Spectrophotometer, Nano-Drop, Wilmington, DE, USA) as well as by Experion (BioRad). Typical RQI values for the purified RNA were 9.3–9.7. Reverse transcription was performed with an input of 1.0 μg, using the iScript cDNA synthesis kit (Bio-Rad) on an iCycler with an IQ5 optical system (Bio-Rad), according to the manufacturer’s protocol. qPCR was performed in technical duplicates, using iTaqTM Universal SYBR Green Supermix (Bio-Rad) on a CFX96 Touch Real-Time PCR Detection System (Bio-Rad) as previously described [43]. Each well in the reaction contained 1× concentration SYBR Green Supermix, 10 ng cDNA, and 12 µmol/L primers for a 20 µL total volume. The primers for qPCR were designed using NCBI/Primer-BLAST and purchased from Integrated DNA Technologies (Coralville, IA, USA) or Bio-Rad (Table 3). The PCR products for all primer pairs were confirmed by agarose gel electrophoresis. The cycle for qPCR was an initial 95 °C for 2 min, followed by 39 repeated cycles of 95 °C 5 s, 53 °C 30 s, and 70 °C 30 s. The cycles were followed by a melt curve assay to ensure the purity of the products. The data were normalized to the reference genes GAPDH, and the relative gene expression calculated with the ΔΔCq method using CFX Maestro software, 2.0 (Bio-Rad) [35,43]. Graphs of the PCR data show the log_2_ relative expression.

### 4.7. Statistics

Statistical analyses were performed using SPSS Statistics, Version 22 software and GraphPad Prism 7 (San Diego, CA, USA). For the comparison of data with time courses with radiation ± captopril treatment, a two-way ANOVA with Tukey’s post-testing was performed to determine the effects of time and treatments for RT-qPCR, Western blotting, GI measurements, and alpha diversity analysis. A permutational analysis of variance (PERMANOVA) was performed for beta diversity metrics. Statistical significance was accepted when *p* < 0.05. For separate studies that performed comparisons of sham-irradiated + captopril to irradiated + captopril groups in the Western blot and qPCR data, unpaired *t* tests were performed.

## Figures and Tables

**Figure 1 ijms-25-04535-f001:**
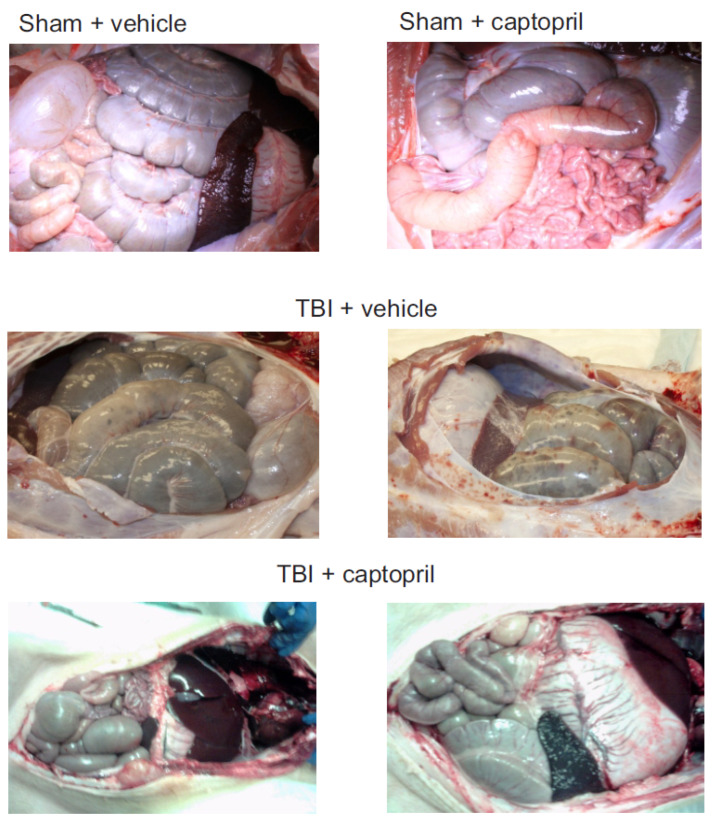
Gross pathology of gastrointestinal tissues in sham-irradiated and irradiated minipigs. Göttingen minipigs were sham-irradiated or exposed to total-body ^60^Co. Minipigs received either vehicle or captopril orally twice daily from 4 hours post-irradiation through to 12 days post-irradiation. Tissues were obtained after euthanasia at 35 days post-irradiation. Representative images are shown.

**Figure 2 ijms-25-04535-f002:**
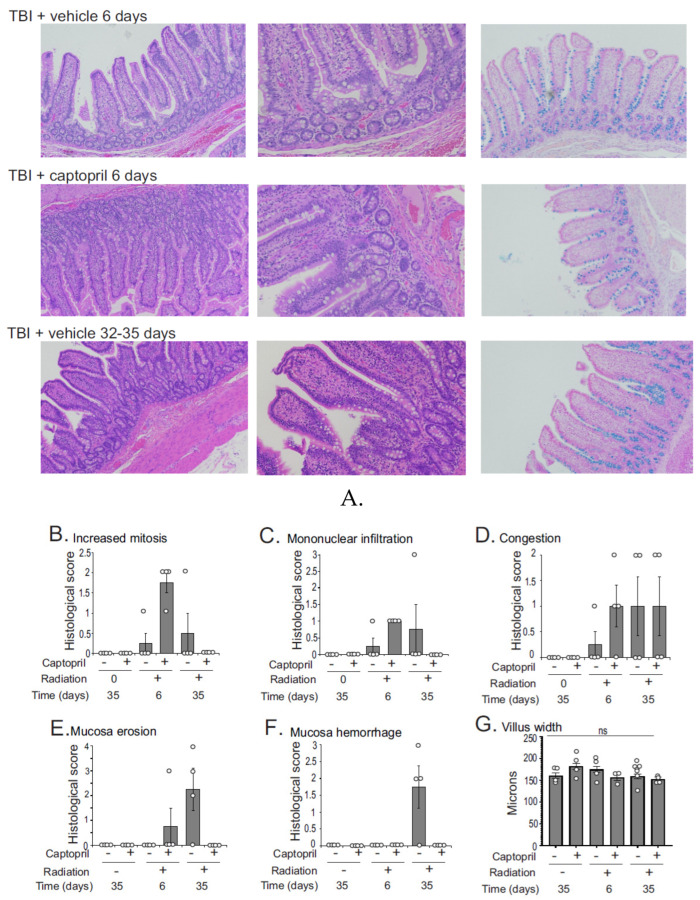
Effect of captopril on histological changes in the intestine following TBI. Göttingen minipigs were sham-irradiated or exposed to total-body ^60^Co. Minipigs received either vehicle or captopril orally twice daily from 4 hours post-irradiation or sham irradiation for 12 days. Tissues were obtained after euthanasia at 6 or 35 days. Sham-irradiated samples were obtained at 35 days. (**A**) Hematoxylin and eosin (H&E) staining of the intestine showing the villi, left panel (10× magnification), and the crypt and intestinal wall region, middle panel (20× magnification); and Alcian blue staining showing goblet cell organization on the villi (10× magnification). Pathological alterations were evaluated using a semi-quantitative scoring system as described in Section 4: (**B**) increased mitosis; (**C**) mononuclear infiltration of the lamina propria; (**D**) vascular congestion; (**E**) mucosa erosion; and (**F**) mucosa hemorrhage. Measurements of histological features: (**G**) villus width; (**H**) villus length; (**I**) crypt depth; (**J**) intestinal wall thickness; and (**K**) percent of goblet cells in the upper 1/3 of villi. At least 10 measurements were obtained per animal. Groups were as follows: sham + vehicle (*n* = 4); sham + captopril (*n* = 4); radiation + vehicle 6 days (*n* = 4); radiation + captopril 6 days (*n* = 3); radiation + vehicle 35 days (*n* = 7); radiation + captopril 35 days (*n* = 4). For goblet cell measurements, *n* = 3 for all groups, except *n* = 4 for sham groups. Graphs show means ± SEM; ns, not significant. White circles indicate individual animal measurements. Data were analyzed by two-way ANOVA and Tukey’s post hoc analysis.

**Figure 3 ijms-25-04535-f003:**
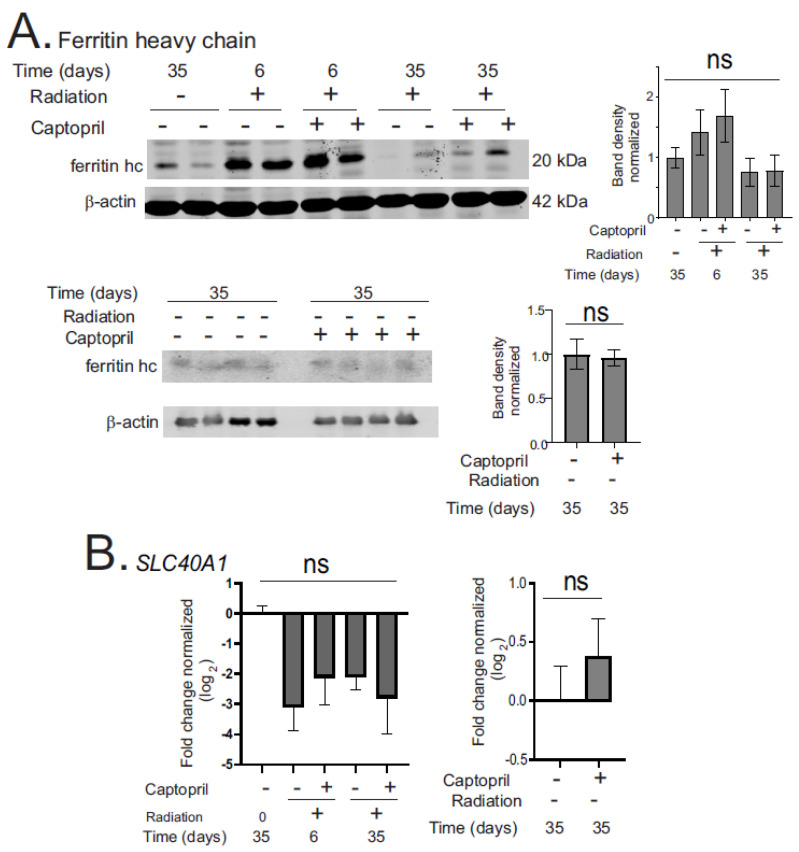
Effect of captopril treatment on iron-binding proteins in the intestine following TBI. Göttingen minipigs were sham-irradiated or exposed to total-body ^60^Co. Minipigs received either vehicle or captopril orally twice daily, starting 4 hours post-irradiation for 12 days. Tissues were obtained after euthanasia at 6 days or 35 days post-irradiation. Sham-irradiated tissues were obtained at 35 days. All samples are *n* = 3 or 4 animals per group. (**A**) Western blot of ferritin heavy chain. Representative results are shown. Bar graphs of band densities normalized to β-actin show means. Graphs show means ± SEM. (**B**) RT-qPCR of *SLC40A1*. Graphs show means ± SEM of log_2_ gene expression normalized to *GAPDH*. For all graphs, ns indicates not significant. Data with radiation time courses were analyzed by two-way ANOVA and Tukey’s post hoc analysis. Data comparing sham + vehicle and sham + captopril were analyzed by unpaired *t* tests.

**Figure 4 ijms-25-04535-f004:**
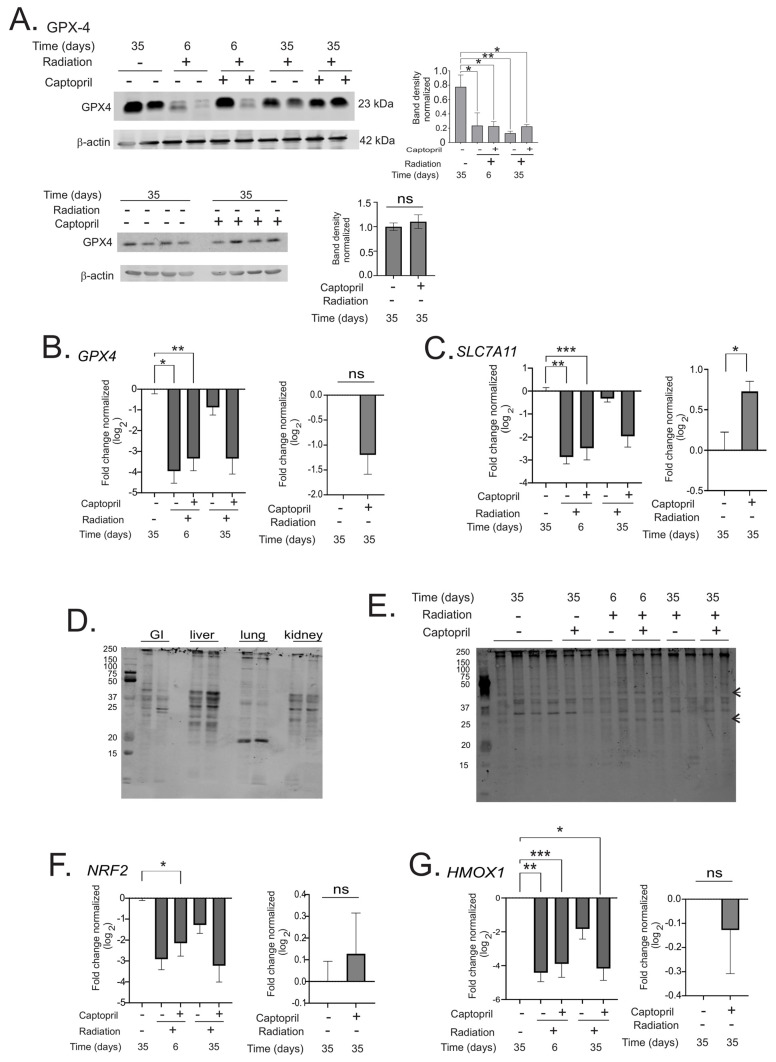
Effect of captopril treatment on ferroptosis in the intestine following TBI. Göttingen minipigs were sham-irradiated or exposed to total-body ^60^Co. Minipigs received either vehicle or captopril orally twice daily starting 4 hours post-irradiation for 12 days. Tissues were obtained after euthanasia at 6 days or 35 days post-irradiation. Sham-irradiated tissues were obtained at 35 days. All samples are *n* = 3 or 4 animals per group. (**A**) Western blot for GPX-4 protein in GI. Representative results are shown. Bar graphs of band densities normalized to β-actin show means. Graphs show means ± SEM. (**B**) RT-qPCR of *GPX4* in GI. (**C**) RT-qPCR of *SLC7A11* in GI. (**D**) Western blot of MDA in intestine (GI), liver, lung, and kidney, sham irradiation. A representative blot is shown. Equal amounts of protein were loaded in each lane. (**E**) Western blot of MDA in GI tissue ± radiation, ±captopril. Equal amounts of protein were loaded in each lane. Arrows indicated bands with increased density after radiation. A representative blot is shown. (**F**) RT-qPCR of *NRF2* in GI. (**G**) RT-qPCR of *HMOX1* in GI. RT-qPCR graphs show means ± SEM of log_2_ gene expression normalized to *GAPDH*, *n* = 4. For all graphs, ns indicates not significant, * indicates *p* < 0.05; ** *p* < 0.01; *** *p* < 0.005. Data for MDA band densities were analyzed using one-way ANOVA. Data with radiation time courses were analyzed by two-way ANOVA and Tukey’s post hoc analysis. Data comparing sham + vehicle and sham + captopril were analyzed by unpaired *t* tests.

**Figure 5 ijms-25-04535-f005:**
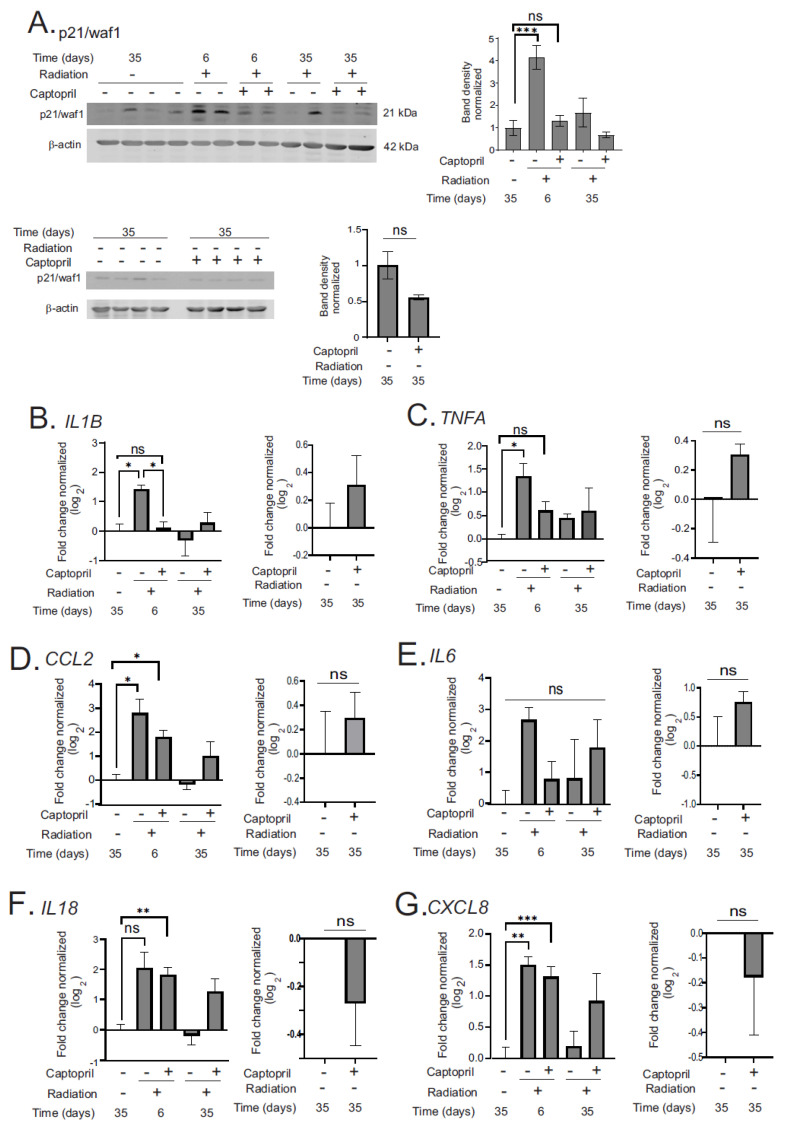
Effect of delayed captopril treatment on pro- and anti-inflammatory cytokine expression in the intestine after TBI. Göttingen minipigs were sham-irradiated or exposed to total-body ^60^Co. Minipigs received either vehicle or captopril orally twice daily, starting 4 hours post-irradiation for 12 days. Tissues were obtained after euthanasia at 6 or 35 days post-irradiation. Sham-irradiated tissues were obtained at 35 days. All samples are *n* = 3 or 4 animals per group. (**A**) Western blot of p21/waf1. Representative results are shown. Bar graphs of band densities normalized to β-actin show means. Graph shows means ± SEM. (**B**–**I**) RT-qPCR was performed on mRNA from the tissues: (**B**) *IL1B*; (**C**) *TNFA1*; (**D**) *CCL2*; (**E**) *IL6*; (**F**) *IL18*; (**G**) *CXCL8*; (**H**) *NLRP3*; and (**I**) *IL10*. RT-qPCR graphs show means ± SEM of log_2_ gene expression normalized to *GAPDH*. For all graphs, ns = not significant; * indicates *p* < 0.05; ** *p* < 0.01; *** *p* < 0.005; **** *p* < 0.001. Data with radiation time courses were analyzed by two-way ANOVA and Tukey’s post hoc analysis. Data comparing sham + vehicle and sham + captopril were analyzed by unpaired *t* tests.

**Figure 6 ijms-25-04535-f006:**
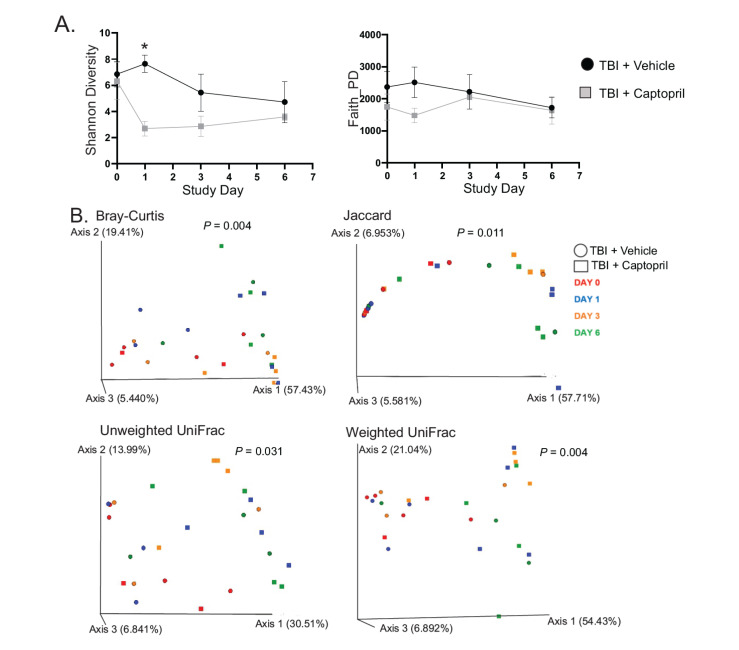
Effect of delayed captopril treatment on GI microbiome following TBI. Göttingen minipigs, 4–6 months of age, were exposed to total-body ^60^Co irradiation or sham-irradiated. Minipigs received either vehicle or captopril twice daily, starting at 4 hours, for 12 days. Anal swabs were obtained at the indicated time points and analyzed, with n = 4 animals per group. (**A**) Impact on Shannon and Faith phylogenetic diversity. Alpha diversity values are represented as mean ± SEM. Two-way ANOVA with post hoc testing; * indicates *p* < 0.05. (**B**) Longitudinal beta diversity metrics evaluating captopril administration following radiation, compared to radiation exposure alone. Symbol shapes depict cohort (TBI + vehicle vs. TBI + captopril), and colors represent different time points. Each data point represents a sample. Permutational analysis of variance (PERMANOVA) results for each metric, comparing the TBI + vehicle versus TBI + captopril, are represented on each PCA plot. Each axis is a principal component showing the percentile describing the proportion of difference in the data set. (**C**) Representation of bacterial taxonomic classifications among groups. Stack bar graphs show the mean relative abundance of the phyla and genus levels for rectal flora. Phyla consisting of >1% of total bacterial composition and the top 10 genera represented, based on baseline organisms.

**Table 1 ijms-25-04535-t001:** Time course of rectal bacteria phyla diversity from Göttingen minipigs following total-body irradiation, with vehicle or captopril treatment.

		Vehicle Group (% of Phyla)				Captopril Group (% of Phyla)			
	Day ^a^	0	1	3	6	0	1	3	6
Phylum									
Firmicutes		50.97 (9.49)	46.11 (2.06) (*p* = 0.99) ^c^	53.10 (11.82) (*p* = 0.99) ^c^	30.96 (10.72) (*p* = 0.99) ^c^	49.79 (3.34) (*p* = 0.99) ^b^	26.66 (4.14) (*p* = 0.22) ^b^ (*p* = 0.17) ^c^	37.60 (10.89) (*p* = 0.82) ^b^ (*p* = 0.24) ^c^	39.61 (5.46) (*p* = 1) ^b^ (*p* = 1) ^c^
Bacteriodetes		2.46 (5.57)	21.15 (4.39) (*p =* 0.99) ^c^	21.87 (9.49) (*p =* 0.99) ^c^	27.41 (9.26) (*p =* 0.99) ^c^	26.13 (2.33) (*p =* 0.99) ^b^	19.07 (1.02) (*p =* 0.09) ^b^ (*p =* 0.30) ^c^	14.58 (2.39) (*p =* 0.71) ^b^ (*p =* 0.10) ^c^	25.52 (6.02) (*p =* 1) ^b^ (*p =* 1) ^c^
Proteobacteria		16.14 (3.53)	20.40 (1.88) (*p =* 0.99) ^c^	17.46 (4.31) (*p =* 0.99) ^c^	34.52 (17.03) (*p =* 0.99) ^c^	12.95 (1.78) (*p =* 0.99) ^b^	48.08 (6.72) (*p =* 0.23) ^b^ (*p =* 0.27) ^c^	39.22 (12.12) (*p =* 1) ^b^ (*p =* 0.94) ^c^	28.51 (4.56) (*p =* 1) ^b^ (*p =* 0.27) ^c^
Actinobacteria		7.29 (0.83)	6.16 (0.29) (*p =* 0.99) ^c^	4.52 (2.22) (*p =* 0.99) ^c^	5.31 (2.38) (*p =* 0.99) ^c^	8.15 (0.65) (*p =* 0.99) ^b^	3.05 (1.64) (*p =* 0.44) ^b^ (*p =* 0.06) ^c^	5.14 (2.83) (*p =* 1) ^b^ (*p =* 0.14) ^c^	4.54 (2.37) (*p =* 1) ^b^ (*p =* 0.23) ^c^
Verrucomicrobia		0.371 (0.12)	3.69 (1.96) (*p =* 0.99) ^c^	0.90 (0.18) (*p =* 0.99) ^c^	0.29 (0.23) (*p =* 0.99) ^c^	0.54 (0.43) (*p =* 0.99) ^b^	0.23 (0.14) (*p =* 0.003) ^b^ (*p =* 0.19) ^c^	0.15 (0.11) (*p =* 0.49) ^b^ (*p =* 0.19) ^c^	0.35 (0.21) (*p =* 1) ^b^ (*p =* 1) ^c^

Relative abundance of phyla longitudinally between groups with standard error of the mean in parenthesis. Differential abundance analysis performed in DESeq2 was used to evaluate the changes in phyla between groups and over time. Animals were exposed to 1.80 Gy (0.5 Gy/min) total body irradiation. Captopril or vehicle (yogurt) treatments were initiated 4 h post-irradiation and administered twice daily. Day 0 is baseline, prior to radiation. ^a^ Days post-irradiation. ^b^ Difference between vehicle- and captopril-treated animals at the same time point, n = 4 animals per time point. ^c^ Difference from day 0 of the same treatment group.

**Table 2 ijms-25-04535-t002:** Time course of rectal bacterial genus-level diversity from Göttingen minipigs following total body irradiation, with vehicle or captopril treatment.

		Vehicle (% of Total Genera)				Captopril (% of Total Genera)			
	Day ^a^	0	1	3	6	0	1	3	6
Genus									
*Lactobacillus*		16.61 (5.51)	8.95 (2.39)	24.09 (12.50)	3.49 (2.86)	22.79 (6.39) (*p* = 0.99) ^b^	0.64 (0.34) (*p* = 0.016) ^b^	10.00 (7.20) (*p* = 0.99) ^b^	1.95 (1.42) (*p* = 0.99) ^b^
*Streptococcus*		1.77 (0.40)	1.64 (1.13)	0.84 (0.21)	0.92 (0.52)	1.79 (0.08) (*p* = 0.99) ^b^	0.81 (0.60) (*p* = 0.88) ^b^	0.81 (0.25) (*p* = 0.99) ^b^	1.10 (0.44) (*p* = 0.99) ^b^
*Enterococcus*		0.80 (0.33)	2.07 (1.10)	2.94 (2.31)	8.56 (3.87)	0.76 (0.07) (*p* = 0.87) ^b^	10.86 (3.35) (*p* = 0.59) ^b^	3.91 (2.19) (*p* = 0.99) ^b^	19.28 (3.73) (*p* = 0.99) ^b^
*Paenibacillus*		1.30 (0.29)	1.26 (0.28)	0.89 (0.39)	0.66 (0.57)	1.13 (0.25) (*p* = 0.99) ^b^	0.47 (0.24) (*p* = 0.083) ^b^	1.0 (0.52) (*p* = 0.99) ^b^	0.20 (0.10) (*p* = 0.99) ^b^
*Blautia*		1.71 (0.37)	1.77 (0.46)	1.21 (0.47)	0.88 (0.68)	1.43 (0.09) (*p* = 0.99) ^b^	0.46 (0.17) (*p* = 0.088) ^b^	0.38 (0.24) (*p* = 0.99) ^b^	0.65 (0.21) (*p* = 0.99) ^b^
*Roseburia*		1.73 (0.49)	1.36 (0.34)	1.10 (0.63)	0.73 (0.66)	1.17 (0.44) (*p* = 0.99) ^b^	0.47 (0.26) (*p* = 0.15) ^b^	0.28 (0.21) (*p* = 0.99) ^b^	0.11 (0.08) (*p* = 0.99) ^b^
f *Lachnospiraceae*		1.34 (0.35)	1.27 (0.33)	0.94 (0.45)	0.68 (0.60)	0.95 (0.22) (*p* = 0.99) ^b^	0.29 (0.11) (*p* = 0.049) ^b^	0.25 (0.18) (*p* = 0.99) ^b^	0.12 (0.08) (*p* = 0.99) ^b^
*Faecalibacterium*		2.27 (0.54)	2.93 (0.61)	1.55 (0.84)	0.84 (0.76)	1.41 (0.29) (*p* = 0.99) ^b^	0.20 (0.07) (*p* = 0.049) ^b^	0.40 (0.33) (*p* = 0.99) ^b^	0.16 (0.12) (*p* = 0.99) ^b^
*Clostridium*		1.61 (0.26)	1.36 (0.23)	1.68 (0.17)	1.06 (0.40)	1.33 (0.27) (*p* = 0.99) ^b^	2.44 (0.94) (*p* = 0.18) ^b^	1.77 (0.35) (*p* = 0.99) ^b^	0.94 (0.20) (*p* = 0.99) ^b^
*Corynebacterium*		1.548 (0.71)	0.32 (0.03)	0.18 (0.05)	2.04 (1.53)	2.16 (0.13) (*p* = 0.99) ^b^	1.08 (0.77) (*p* = 0.16) ^b^	0.60 (0.26) (*p* = 0.99) ^b^	3.20 (1.87) (*p* = 0.99) ^b^
*Bacteroides*		10.74 (6.26)	7.91 (2.45)	13.29 (10.27)	21.49 (10.81)	10.35(4.74) (*p* = 0.99) ^b^	11.81 (2.23) (*p* = 0.58) ^b^	7.96 (2.66) (*p* = 0.99) ^b^	21.47 (5.55) (*p* = 0.99) ^b^
*Prevotella*		2.79 (0.77)	3.42 (1.33)	1.59 (0.77)	1.20 (1.11)	2.98 (0.24) (*p* = 0.99) ^b^	0.32 (0.11) (*p* = 0.045) ^b^	0.53 (0.44) (*p* = 0.99) ^b^	0.36 (0.21) (*p* = 0.99) ^b^
*Phocaeiola*		2.71 (0.40)	4.24 (0.88)	3.18 (0.61)	1.67 (1.14)	5.50 (0.71) (*p* = 0.99) ^b^	2.36 (1.12) (*p* = 0.62) ^b^	1.45 (1.24) (*p* = 0.99) ^b^	1.32 (0.51) (*p* = 0.99) ^b^
f *Pastearellaceae*		1.99 (0.87)	1.09 (0.74)	3.77 (2.53)	5.34 (3.41)	2.75 (0.80) (*p* = 0.99) ^b^	2.13 (1.24) (*p* = 0.31) ^b^	8.04 (4.51) (*p* = 0.99) ^b^	7.42 (2.00) (*p* = 0.99) ^b^
*Escherichia*		3.76 (3.31)	8.32 (2.87)	6.27 (3.40)	22.11 (19.60)	1.47 (0.94) (*p* = 0.99) ^b^	33.42 10.31) (*p* = 0.30 )^b^	19.34 10.79) (*p* = 0.99) ^b^	15.75 (5.01) (*p* = 0.99) ^b^
*Ruminococcus*		1.22 (0.42)	1.08 (0.20)	0.72 (0.36)	0.48 (0.44)	0.77 (0.06) (*p* = 0.99) ^b^	0.12 (0.04) (*p* = 0.053) ^b^	0.26 (0.22) (*p* = 0.99) ^b^	0.12 (0.09) (*p* = 0.99) ^b^

Relative abundance of phyla longitudinally between groups with standard error of the mean in parenthesis. Differential abundance analysis performed in DESeq2 was used to evaluate the changes in genera between groups and over time. Animals were exposed to 1.80 Gy (0.5 Gy/min) total body irradiation. Captopril or vehicle (yogurt) treatments were initiated 4 h post-irradiation and administered twice daily. ^a^ Days post-irradiation. ^b^ Difference between vehicle- and captopril-treated animals at the same time point, n = 4 animals per time point.

**Table 3 ijms-25-04535-t003:** Primers for qPCR.

		Forward Primer	Reverse Primer
Gene (Amplicon Size)			
*CCL2* (121 bp)		5′-AGAAGATCTCGATGCAGCGG-3′	5′-TTCTGCTTGGGTTCTGCACA-3′
*CXCL8* (166 bp)		5′-TGCAGAACTTCGATGCCAGT-3′	5′-CCACTTTTCCTTGGGGTCCA-3′
*GAPDH* (196 bp)		5′-GTCGGAGTGAACGGATTTG-3′	5′-CATTGATGACAAGCTTCCCG-3′
*GPX4* (252 bp)		5′-GAATTCGGCACGAGAGGAGC-3′	5′-TTGGTGACGATGCACACGTA-3′
*HMOX1* (277 bp)		5′-CGCCTTCCTGCTCAACATTC-3′	5′-ACGGTTGCATTCACAGGGTT-3′
*IL1B* (162 bp)		5′-TGTCTGTGATGCCAACGTG-3′	5′-TCATCTCCTTGCACAAAGCTC-3′
*IL6* (101 bp)		5′-GTCGAGGCCGTGCAGATTAG-3′	5′-GCATTTGTGGTGGGGTTAGG-3′
*IL10* (not disclosed)		Prime PCR assay, Bio-Rad primer	
*IL18* (147 bp)		5′-GGCAGTAACCATCTCTGTGCA-3′	5′-TGTCCAGGAACACTTCTCTGA-3′
*NLRP3* (70 bp)		5′-TTCTTCCATGGCTCAGGACAC-3′	5′-AGGGCATAGGTCCACACAAAA-3′
*NRF2* (268 bp)		5′-CTAAGGGTGCTCCTTTGCGA-3′	5′-CATGCTCCTTCCGTCGTTGA-3′
*PTGS2* (281 bp)		5′-AGGACCAGCTTTCACCAAAGG-3	5′-TATGTTCCCGCAGCCAGATTG-3
*SLC7A11* (287 bp)		5′-CCTGGGCAGGAGAAAGTTGT-3′	5′-CAGACTCGCACAAAAGCTGG-3′
*SLC40A1* (268 bp)		5′-TAAAGTGGCCCAGACGTCAC-3′	5′-TCGTATTGTAGCATTCATATCTGC-3′
*TNFA* (129 bp)		5′-GATTCAGGGATGTGTGGCCT-3′	5′-GCCACATTCCAGATGTCCCA-3′

*Sus scrofa* (pig) sequences for the following: *CCL2* (chemokine [C-C motif] ligand 2; gene ID 397422), *CXCL8* (chemokine [C-X-C motif] ligand 8 or interleukin 8; gene ID 396880), *GAPDH* (glyceraldehyde-3-phosphate dehydrogenase; gene ID 396823), *GPX4* (glutathione peroxidase 4; gene ID 399537), *HMOX1* (hemoxygenase 1; gene ID 445512), *IL1B* (interleukin 1 beta; gene ID 397122), *IL6* (interleukin 6; gene ID 399500), *IL10* (interleukin 10; gene ID 397106), *IL18* (interleukin 18; gene ID 387057), *NLRP3* (NACHT, LRR and PYD domains-containing protein 3 or cryopyrin; gene ID 100514823), *NRF2* (nuclear factor, erythroid 2 like 2, NFE2L2;100516343), *SLC7A11* (solute carrier family 7 member A11; gene ID 100623148), *SLC40A1* (solute carrier family 40 member A1; gene ID 100737517), and *TNFA* (tumor necrosis factor alpha; gene ID 397086).

## Data Availability

Data for the qPCR, immunohistochemistry, and Western blot results can be obtained from the corresponding author, Dr. Regina Day, by request at regina.day@usuhs.edu. Data related to the microbiome study are available from Dr. David Burmeister by request at david.burmeister@usuhs.edu.

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
