# Peer review of "Ferroptosis, Inflammation, and Microbiome Alterations in the Intestine in the Göttingen Minipig Model of Hematopoietic-Acute Radiation Syndrome"

_ijms, 2024, doi:10.3390/ijms25084535_

Round 1
Reviewer 1 Report (New Reviewer)
Comments and Suggestions for Authors
1. General comments
The aim of the study is to elucidate the mechanisms by which captopril suppresses hematopoietic acute radiation syndrome (H-ARS) with damage to multiple organ systems following total body irradiation (TBI).
It is interesting to note that captopril was found to have an effect on the diversity of the rectal microbiota after irradiation with changes over time. The description of this statistical analysis should be better described so that the methodology can be reproduced.
2. Line 25, “60Co”
60Co should be 60Co.
3. Line 333, “PERMANOVA”
It is better to spell it out such as permutational multivariate analysis of variance.
4. Line 681, “Data Availability Statement”
Example sentences are given verbatim. They must be replaced by the explanations in this study.
5. Table 1, “bDifference between vehicle- and captopril treated animals at the same time point, n=4 animals per time point.”
On day 6 all phyla show p=1, is this correct?
6. Table 1, “cDifference from Day 0 of the same treatment group.”
Even on day 6, all phyla show p=0.99, is this true?
7. Figure 6
Is it confusing to use periods to separate the differences between graphs A, B and C? Is it intentional that the first word in the first sentence of the description of graph A is lower case? In graph B, it would it be better to state clearly which (multi-dimensional) scale each axis is on as beta diversity.
Author Response
Revisions for manuscript “Ferroptosis, Inflammation, and Microbiome Alterations in the Intestine in the Göttingen Minipig Model of Hematopoietic-Acute Radiation Syndrome” for consideration for publication in the Special Issue “Regulation and Targeting of Ferroptosis in Tumor and Beyond” in International Journal of Molecular Sciences.
We thank the Editor and the Reviewers for the opportunity to revise our manuscript for reconsideration for publication. Below is a point-by-point response to the Reviewers’ comments.
Reviewer 1:
- General comments
The aim of the study is to elucidate the mechanisms by which captopril suppresses hematopoietic acute radiation syndrome (H-ARS) with damage to multiple organ systems following total body irradiation (TBI). It is interesting to note that captopril was found to have an effect on the diversity of the rectal microbiota after irradiation with changes over time. The description of this statistical analysis should be better described so that the methodology can be reproduced.
The following sentence is now included in section 4.4. Microbiome which should allow one to reproduce the analysis: "The Biom table was imported into Quantitative Insights into Microbial Ecology (QIIME2, version 2021.8) for generation of alpha and beta diversity metrics [97]". Statistical tools used for alpha and beta diversity were added or clarified in the methods section 4.7 Statistics. We have added sampling depth of 485000 for diversity calculations in the Microbiome portion of the methods.
- Line 25, “60Co” 60Co should be 60Co.
This correction has been made.
- Line 333, “PERMANOVA” It is better to spell it out such as permutational multivariate analysis of variance.
This correction has been made.
- Line 681, “Data Availability Statement” Example sentences are given verbatim. They must be replaced by the explanations in this study.
This correction has been made. The data will be made available to researchers upon request to the authors.
- Table 1, “bDifference between vehicle- and captopril treated animals at the same time point, n=4 animals per time point.” On day 6 all phyla show p=1, is this correct?
Yes, DESeq2 results indicate p=1 at day 6 for differences between vehicle-captopril groups. This may stem from our small sample size which is noted as a limitation in the discussion.
- Table 1, “cDifference from Day 0 of the same treatment group.” Even on day 6, all phyla show p=0.99, is this true? This is correct as indicated with Proteobacteria and Actinobacteria showing pvalues of 0.27 and 0.23, respectively. Again, as mentioned above, sample size may play a role here.
- Figure 6 Is it confusing to use periods to separate the differences between graphs A, B and C? Is it intentional that the first word in the first sentence of the description of graph A is lower case? In graph B, it would it be better to state clearly which (multi-dimensional) scale each axis is on as beta diversity.
Agreed, we have updated the figure legend to include A), B), and C) instead of periods to separate panel letters. The first word for graph A has been capitalized. We have clarified the meaning of each axis which is a principal component. PCA figures find a new coordinate system in which every point has a new (x,y) value. The axes don't actually mean anything physical; they're combinations of height and weight called "principal components" that are chosen to give one axes lots of variation.
Reviewer 2 Report (New Reviewer)
Comments and Suggestions for Authors
In general, this study seems relevant and provides interesting data about possible mechanisms how Captopril alleviates TBI pathology in the gut. Studying such questions in non-rodent animal models is challenging and relevant. There are some concerns:
General points:
- The link from Captopril treatment with ferroptosis is not entirely clear. Some non-significant findings appear to trend in interesting directions, but the overall contribution of ferroptosis to the Captopril treatment remains a little shaky.
- Overall, it would be better to depict individual datapoints in addition to bar graphs. Especially in Figure 2, it would be good to see how many data points are “0”.
- Perhaps the most interesting finding to me is the difference in microbiota composition. As the authors note, the lack of a Captopril only group is a limitation of the study. If at all possible, this data would make the paper much stronger.
- It would be good to include the used statistical tests in all Figure legends.
- The discussion is very long and recounts the findings from the result section a lot. It could be more concise.
- Thorough mechanistic studies are hard in non-mouse models. It would, however, be helpful if the discussion proposed a clearer hypothesis about how the different results are linked. It appears to me that maybe the most likely hypothesis is that Captopril affects the microbiota composition, which in turn leads to differences in inflammation and differences in ferroptosis markers are downstream of this difference. This or alternative hypotheses could be discussed in a clearer way.
- Are the data from the microbiota analysis matchable to disease severity and data in the other figures? Correlation analyses of microbiota to disease outcome would strengthen the potential mechanism described above. Is there a given microbiota composition that correlates with protection from inflammation in the Captopril group?
- I appreciate the very careful framing and that the authors do not overinterpret the data. Given that most of the ferroptosis part is not significant, the authors could condense this part slightly, especially if it is possible to include a figure linking microbiota differences to other markers in correlation analyses as suggested above.
Minor points:
- If 50% vs 25% of animals have ecchymoses, that seems relevant. It would be good to include quantified data in the figure next to the pictures in Fig. 1
- 2F) The statistics are not always clear to me. Why is the data from 2F not significant and in what test?
- There is a scale on the bottom of Fig. 2 (cont) that belongs to no data.
- Figure 4: It appears the Figure references in the text are mixed up. E.g. the text states that I can find the GPC-4 data in 4A and not 4C if I don’t misunderstand.
- Figure 5A: Also here, a sham + Captopril group would be very helpful.
- Figure 5B)-5I) Significances for day 6 between Captopril and control should be indicated rather than only in 5A). What statistical test was used? This should be in the figure legend. How many data points are there? Again, it would be better to show the actual values in addition to bar graphs.
Author Response
Revisions for manuscript “Ferroptosis, Inflammation, and Microbiome Alterations in the Intestine in the Göttingen Minipig Model of Hematopoietic-Acute Radiation Syndrome” for consideration for publication in the Special Issue “Regulation and Targeting of Ferroptosis in Tumor and Beyond” in International Journal of Molecular Sciences.
We thank the Editor and the Reviewers for the opportunity to revise our manuscript for reconsideration for publication. Below is a point-by-point response to the Reviewers’ comments.
Reviewer 2:
In general, this study seems relevant and provides interesting data about possible mechanisms how Captopril alleviates TBI pathology in the gut. Studying such questions in non-rodent animal models is challenging and relevant. There are some concerns:
General points:
- The link from Captopril treatment with ferroptosis is not entirely clear. Some non-significant findings appear to trend in interesting directions, but the overall contribution of ferroptosis to the Captopril treatment remains a little shaky.
We agree that Captopril does not modulate all of the effects of radiation in the GI. This study was undertaken not just to elucidate the positive effects of Captopril, but also to determine where Captopril does not have an effect. This might indicate that additional agents may be needed for radiation protection in a mass casualty or military scenario. We have included this in the Discussion.
- Overall, it would be better to depict individual datapoints in addition to bar graphs. Especially in Figure 2, it would be good to see how many data points are “0”.
We now show the individual animal averages for Figure 2, so the number of data points at 0 can be observed.
- Perhaps the most interesting finding to me is the difference in microbiota composition. As the authors note, the lack of a Captopril only group is a limitation of the study. If at all possible, this data would make the paper much stronger.
Unfortunately, we do not currently have funding to purchase more animals for this study. We have included additional comments in the Discussion regarding this.
- It would be good to include the used statistical tests in all Figure legends.
Statistical information is now included in all figure legends.
- The discussion is very long and recounts the findings from the result section a lot. It could be more concise.
We have shortened the Discussion to remove repetition of results where possible.
- Thorough mechanistic studies are hard in non-mouse models. It would, however, be helpful if the discussion proposed a clearer hypothesis about how the different results are linked. It appears to me that maybe the most likely hypothesis is that Captopril affects the microbiota composition, which in turn leads to differences in inflammation and differences in ferroptosis markers are downstream of this difference. This or alternative hypotheses could be discussed in a clearer way.
We have included a clearer conclusion in the Discussion regarding the effects of Captopril, where our findings in the minipig are consistent with our findings in our murine model. At this time, because of the limited data on the effects of Captopril on the microbiome, we feel that we need more data to make a firm conclusion.
- Are the data from the microbiota analysis matchable to disease severity and data in the other figures? Correlation analyses of microbiota to disease outcome would strengthen the potential mechanism described above. Is there a given microbiota composition that correlates with protection from inflammation in the Captopril group?
We do not have sufficient data to match the data from the microbiome to the individual data points from the histology since the samples were pooled from the microbiome data. A larger study would be required for this analysis.
- I appreciate the very careful framing and that the authors do not overinterpret the data. Given that most of the ferroptosis part is not significant, the authors could condense this part slightly, especially if it is possible to include a figure linking microbiota differences to other markers in correlation analyses as suggested above.
We have tried to reduce the discussion of ferroptosis in the Discussion.
Minor points:
- If 50% vs 25% of animals have ecchymoses, that seems relevant. It would be good to include quantified data in the figure next to the pictures in Fig. 1
Since there were only 4 animals in the study, and the observations were made subjectively, we can’t provide statistical analysis for this.
- 2F) The statistics are not always clear to me. Why is the data from 2F not significant and in what test?
For many experiments, we performed two sets of experiments. In the first set, we compared the time course of radiation ± captopril, compared with sham without captopril; these data sets were analyzed using two-way ANOVA and Tukey’s post-test. In the second set of experiments, we compared just captopril administration to control, without radiation. These data sets used unpaired T test analysis. The statistical analysis for the microbiome is also now described in the Statistics section of the Methods. We added alpha and beta diversity metric calculation information.
- There is a scale on the bottom of Fig. 2 (cont) that belongs to no data.
This has been corrected.
- Figure 4: It appears the Figure references in the text are mixed up. E.g. the text states that I can find the GPX-4 data in 4A and not 4C if I don’t misunderstand.
We have double checked the labels for Fig 4 in the figure legend. The figure legend is now corrected.
- Figure 5A: Also here, a sham + Captopril group would be very helpful.
We agree that additional data sets would be preferable. If we receive new funding for our studies, we will be able to include more animal groups.
- Figure 5B)-5I) Significances for day 6 between Captopril and control should be indicated rather than only in 5A). What statistical test was used? This should be in the figure legend. How many data points are there? Again, it would be better to show the actual values in addition to bar graphs.
We performed two separate sets of experiments for our manuscript.
In our first set of experiments, we conducted our western blots and qPCR assays in groups that included sham irradiated vehicle controls and all irradiated groups +/- captopril. These data were analyzed by two-way ANOVA with Tukey’s posttest.
At a later time, we were asked to compare sham irradiated captopril to sham irradiated vehicle animals, which were all euthanized at 35 days. We performed separate experiments to compare control and day 35 captopril without radiation; these are depicted in separate bar graphs and separate western blots, as shown in Figure 5. Because these experiments were performed separately, and at different times, we felt it was better to show the data separately. These data were analyzed by unpaired T tests.
Figures 1B-K show tissue measurements, and we were able include all the data in one single plot despite performing measurements at different times. These data were analyzed by two-way ANOVA with Tukey’s posttest.
This manuscript is a resubmission of an earlier submission. The following is a list of the peer review reports and author responses from that submission.
Round 1
Reviewer 1 Report
Comments and Suggestions for Authors
In this manuscript, the authors explore intestinal inflammation, ferroptosis, and microbiome changes in H-ARS mice. Unfortunately, I feel that the biggest problem with this study is that the sections are not well connected and each section is not studied in depth.
1. in Figure 1, I would recommend giving a photo of the intestines as well as a more detailed pathology score.
2. in Figure 2, it is not sufficient to determine the extent of ferroptosis occurring from the expression of only a few genes. I suggest evaluating the lipid peroxidation of intestinal tissue and mitochondrial morphology of intestinal epithelial cells.
3.Figure 3, It is recommended to explore inflammatory cytokine expression changes based on the evaluation of immune cell infiltration in intestinal tissues using immunofluorescence or flow cytometry.
Comments on the Quality of English LanguageIn this manuscript, the authors explore intestinal inflammation, ferroptosis, and microbiome changes in H-ARS mice. Unfortunately, I feel that the biggest problem with this study is that the sections are not well connected and each section is not studied in depth.
1. in Figure 1, I would recommend giving a photo of the intestines as well as a more detailed pathology score.
2. in Figure 2, it is not sufficient to determine the extent of ferroptosis occurring from the expression of only a few genes. I suggest evaluating the lipid peroxidation of intestinal tissue and mitochondrial morphology of intestinal epithelial cells.
3.Figure 3, It is recommended to explore inflammatory cytokine expression changes based on the evaluation of immune cell infiltration in intestinal tissues using immunofluorescence or flow cytometry.
Author Response
We would like to thank the reviewers for their comprehensive evaluation of our manuscript. We have responded to as many of requests for additional data as possible. A point-by-point response to reviewers follows. We hope that our manuscript will now be found acceptable for publication.
Reviewer 1
In this manuscript, the authors explore intestinal inflammation, ferroptosis, and microbiome changes in H-ARS mice. Unfortunately, I feel that the biggest problem with this study is that the sections are not well connected and each section is not studied in depth.
Murine models of radiation biology, including the investigation of total body irradiation and localized radiation, have been the most extensively characterized model for the study of radiation biology. Additionally, the murine models are used as the initial preclinical model for radiation countermeasure development. Under the FDA’s Animal Rule, two preclinical models are required for the validation of radiation countermeasures, both of which must recapitulate the reaction of humans and/or human tissue to radiation. One of the models for FDA approval can be a rodent model, but the second animal model must be non-rodent. Historically, radiation countermeasure development has required studies in nonhuman primates (NHPs), but this has changed due to the lack of NHPs for this type of research. The FDA has recently approved the use of swine data for radiation countermeasure development. However, the use of minipigs for these studies is hampered by the limited number of studies performed in swine and the limited analyses of radiation effects.
Given this situation in the field, our study has two goals: to characterize the biological effects of radiation in the minipig ARS model, and to determine the effects of the known radiation countermeasure captopril in this system. Our study was limited by the cost of the experiments, a factor that is common for large animal experiments. We were unable to obtain tissues at all time points for which we previously obtained murine tissues for similar studies. We were also not able to perform these initial experiments in both sexes, and because of a cap on the funding for the project, we proposed only male animals to ensure that we obtained one complete data set. Additionally, there are very limited reagents available for the study of swine tissues, especially a limitation in antibodies for either western blots or immunohistochemistry.
We have included additional explanations of our study as well as additional requested data where possible.
- in Figure 1, I would recommend giving a photo of the intestines as well as a more detailed pathology score.
Previous studies have determined that radiation induces internal hemorrhage by 35 days post-irradiation that can be observed in gross anatomical analysis; these events have previously been scored (Moroni et al., 2011). We have now included photographs of the gross anatomy of the intestines post-irradiation, so that the typical level of hemorrhage can be seen (Figure 1). Our veterinary pathologist performed a detailed analysis of the histological sections; the observational scores are now provided in Figure 2B-D, including increased mitosis, and mononuclear infiltration, congestion and mucosa erosion. The scoring system is described in the Methods section. Note that because these are semi-quantitative measures, it is not possible to perform statistics. Overall, this analysis showed that in the radiation + vehicle groups, there was slight or less than slight increased mitosis, mononuclear infiltration, and congestion; within this group there was also mild erosion and mucosa hemorrhage. In the captopril treated groups, there was also slight or less than slight increased mitosis, mononuclear infiltration, and congestion. There was less evidence of mucosal erosion and mucosal hemorrhage at 35 days post-irradiation compared to the vehicle group. The pathologist determined that over this time course there was no evidence of neutrophilic or lymphocytic infiltration or loss of crypts.
- in Figure 2, it is not sufficient to determine the extent of ferroptosis occurring from the expression of only a few genes. I suggest evaluating the lipid peroxidation of intestinal tissue and mitochondrial morphology of intestinal epithelial cells.
In order to address this issue, we purchased an assay to identify lipid peroxidation that was supposedly applicable for extracts from tissues (Sigma). Although we spent several weeks on this assay, we were unable to obtain any data due to a failure of the assay. Because of the time limitation for our revision, we are not able to perform the requested analysis of mitochondrial morphology, as we do not have expertise in this area. Additionally, we are limited by the 6 day and 35 day time points that we collected for this experiment, which may not include a time point of maximal lipid peroxidation. However, GPX-4 is the sole mammalian enzyme known to reduce phospholipid oxides to phospholipid alcohols, and cystine (imported by SLC7A11) is a required factor for this reduction (Liang et al. 2023). Studies have shown that genetic or pharmacologic inhibition of GPX-4 or SLC7A11 induces ferroptosis (Lei et al. 2020). Therefore, to address this issue, we now include in the Discussion a statement that our finding are consistent with ferroptosis, and that identification of lipid peroxidation and alterations in mitochondrial morphology would be desirable data for a future study.
3.Figure 3, It is recommended to explore inflammatory cytokine expression changes based on the evaluation of immune cell infiltration in intestinal tissues using immunofluorescence or flow cytometry.
Unfortunately, these experiments have ended, and we do not have fresh tissues or correctly fixed tissues available that would be appropriate for flow cytometry. Additionally, we have experienced a great deal of difficulty in identifying antibodies that are effective in our minipig tissues. We have now included the pathologist’s analysis of inflammatory cell infiltration. Mononuclear infiltration is shown in Fig. 2. The pathologist also determined that infiltration of neutrophils and leukocytes was unremarkable in the tissues.
Reviewer 2 Report
Comments and Suggestions for Authors
Dear authors,
please find my comments in the attached file.
Best regards

Comments on the Quality of English LanguageDear authors,
the English language of this manuscript is well understandable.
Besrt regards
Author Response
We would like to thank the reviewers for their comprehensive evaluation of our manuscript. We have responded to as many of requests for additional data as possible. A point-by-point response to reviewers follows. We hope that our manuscript will now be found acceptable for publication.
Reviewer 2
Recommendation: Major revision
The manuscript describes a study of gut alterations in the Göttingen Minipig Model of Hematopoietic-Acute Radiation Syndrome. Male minipigs were total body-exposed to ~1.8 Gy gamma-rays and a subgroup was treated with the ACE inhibitor captopril. Ileum tissue was collected after euthanasia, 6 or 35 days after irradiation. Furthermore, rectal swabs were used for gut microbiome analysis. Tissue analysis included histology, Western blots and real-time RT-RT-qPCR of selected proteins / RNAs involved in ferroptosis or inflammation. TBI seems to induce ferroptosis and p21, and some inflammatory cytokines, in the ileum. Captopril counteracted parts of these effects.
- Due too a low image quality and a formatting issue in the microbiome tables, it is impossible to make a complete judgement of the presented results. The resolution of the figures has to be improved and tables have to be corrected to make this possible.
Improved images are now provided for all histology with the help of the Journal editorial team. We are unsure of the formatting problems with the microbiome tables, since these appeared to be imported into the Word file correctly. We will therefore provide the tables to the journal staff to import correctly into the journals format.
- At this stage, the selection of endpoints (ferroptosis, inflammation, microbiome) appears rather random. Only in the discussion, the possible links become a little bit clearer. A better explanation in the introduction would be helpful.
We now include in the Introduction, in paragraph 2, an explanation of the GI effects of doses of radiation that induce hematopoietic-acute radiation syndrome (H-ARS) in Göttingen minipigs. This includes the time points that were previously examined for GI injury. We also include an explanation of the time points selected for our study in paragraph 4 of the Introduction. Basically, previous studies showed that, following doses of radiation that induce H-ARS, GI injury was observed at 3-6 days post-irradiation. Therefore, our earliest time point examined were 6 days post-irradiation for histological samples, and 3-6 days for microbiome alterations. For our late time point we selected 35 days post-irradiation, the time point which is the gold standard for survival from H-ARS. Our goal was therefore to characterize early and last GI effects of radiation.
We also include this statement in paragraph 2 of the Introduction, to help clarify the goals of our study. “In the Göttingen minipig, subacute GI injury was observed at doses of radiation that induced H-ARS (1.7 – 1.9 Gy, 0.6 Gy/min); symptoms of GI injury (dry stools, anorexia and no weight gain) were observed in most animals between several hours to 14 days post-irradiation [18]. More severe GI injury was observed with 2.2 – 4 Gy (0.6 Gy/min) TBI, a dose of radiation that also is associated with H-ARS but not GI-ARS [19]. Indicators of GI injury at 2.2 – 4 Gy include villar blunting and fusion, intestinal hemorrhages, and increased concentration of serum citrulline (indicator of loss of GI barrier integrity); these alterations were evident from 3-6 days post-irradiation [18, 19]. Signs of acute sickness following H-ARS irradiation (weakness, facial edema, petechia and desquamation in areas of the skin) and initial peripheral blood cell recovery are observed by 35 days post-irradiation [18], but resolution of subacute GI injuries at this late time point has not been completely defined.”
- The microbiome data are difficult to interpret as the data from the mock-irradiated minipigs are missing. It is therefore unclear what the effect of TBI on the gut microbiome is. Captopril seems to have an influence on the gut microbiome of irradiated minipigs, but again, it is impossible to interpret the results as the microbiome data of minipigs treated with captopril only are missing.
Our research study was funded to determine the impact of captopril on early radiation injuries, and unfortunately a sham-irradiated captopril treatment group could not be included in the experiment examining the early time points. In large animal experiments, it is sometimes not possible to include all control groups due to cost, availability of housing, etc. We previously determined that captopril was not toxic at 35 days post-irradiation (Rittase et al., 2021), and studies with captopril over the last 40+ years have not demonstrated GI toxicity of this drug which is commonly used to treat hypertension and heart failure. The reviewer is correct that inclusion of the captopril sham irradiated treatment group would provide additional information regarding the effect of captopril alone on the microbiome. A recent publication on the effect of captopril treatment for 2 weeks had little or no effect on diversity in the gut microbiome in Wistar Kyoto rats (Yang et al. 2019), however no similar studies were performed in swine or in humans. This data is now cited in the Discussion section.
- Furthermore, no information on symptoms of the minipigs are given, especially whether captopril had a positive influence on them.
The characteristics of hematopoietic acute radiation syndrome (H-ARS) are stated in the first paragraph of the Introduction. The symptoms are the same in all animal models of H-ARS, including a loss of mature blood cells in the periphery and loss of hematopoietic progenitors in the bone marrow. We have stated in the Abstract that our laboratory showed that captopril improves survival from hematopoietic acute radiation syndrome, including suppression of radiation-induced acute inflammation. This is also stated in the last paragraph of the Introduction.
More points are listed below.
Abstract
- “1.79 or 1.80 Gy” – these two doses differ by only 0.01 Gy (10 mGy) – why is it important to state that doses with such a minor difference were used?
Minipigs display a steep response curve to changes in the dose of total body irradiation. For example, 1.79 Gy (0.6 Gy/min) is a 50% lethal exposure, but 1.9 Gy is 100% lethal. Because of this steep curve, we felt that it is important to accurately indicate the dose of radiation to two significant figures.
- “p21/waf1 (senescence marker)” => p21WAF1/CIP1 (Gene name CDKN1A) can be involved in senescence, but most importantly, it mediates ionizing radiation-induced cell cycle arrest. p16, senescence-associated beta-galactosidase and the senescence-associated secretory profile are better senescence markers.
Because fresh tissues are not available for SA-βGAL analysis, we are unable to perform this assay. We have performed RT-qPCR to determine local synthesis of pro-inflammatory cytokines, which we show are present in the tissues. Unfortunately, we were unable to find a p16 antibody that would recognize the porcine protein. We have modified this section to state that transient p21waf1 upregulation is consistent with cell cycle arrest post-irradiation while sustained upregulation is consistent with accelerated senescence. We state that our limited time course study is not sufficient to distinguish between these. We included this information in the Discussion as well. We hope to follow up this study in the future to obtain more time points.
- “RT-qPCR analysis” => Do you mean RT-RT-qPCR analysis?
This correction has been made in the text.
- “Our data suggest that captopril mitigates senescence,” => as p21 is a marker for cell cycle arrest, no conclusion on senescence can be made.
We have modified this statement that p21 upregulation is consistent with cell cycle arrest.
Results
- Figure 1: Image quality is too low, especially for B to F. The 6 days untreated control is missing.
Improved images have been supplied and imported into the text with the help of the Journal editorial team.
- B to F: The label “Radiation (Gy)” is confusing if only a “+” is given – the plus would fit to “Radiation”, the actual dose to “Radiation (Gy)”.
This correction has been made in all figures.
- Figure 2: Image quality is too low
Improved images have been supplied and imported into the text with the help of the Journal editorial team.
- “Bar graphs of band densities normalized to ß-actin show means. Graphs show means ± SEM.”
=> “Bar graphs show means ± SEM of band densities that were normalized to ß-actin”.
This correction has been made.
- “RT-qPCR” => “RT-RT-qPCR”
Our experiments were performed by first doing reverse transcription to produce cDNA. The subsequent cDNA was then used for quantitative polymerase chair reactions. This was not done by real-time reverse transcription PCR.
- “normalized to GAPDH”: for RT-RT-qPCR, a set of at least 3 reference genes is recommended.
Otherwise, please show that GAPDH expression was stable under the conditions of this work.
Our laboratory extensively researched the stability of a variety of housekeeping genes to identify genes that did not fluctuate in animal tissues post-irradiation and to ensure that the concentrations of the housekeeping gene mRNAs were similar to levels of the mRNAs of our target genes. Our investigation included GAPDH, β-actin, 18S, Rplp0, HPRT1, Rps29, Rpl4, and Oaz1. We found that many of these were not stable post-irradiation, such as β-actin. Additionally, although Rpl4 was stable, it was orders of magnitude higher than our target genes and could not be used reliably for quantification. We also purchased custom-made PCR plates from BioRad that already included 3 housekeeping genes: GAPDH, HPRT1, and TBP; all of these were highly reproducible between experiments (see below). We found that all of these housekeeping genes were accurate and reproducible between experiments. (See Attachment 2.)
Experiment 1: Pig BM1 |
Experiment 2: Pig BM2 |
|||||
GAPDH |
7 |
18.85 |
GAPDH |
7 |
18.62 |
|
GAPDH |
8 |
19.30 |
GAPDH |
8 |
19.25 |
|
GAPDH |
9 |
22.36 |
GAPDH |
9 |
21.99 |
|
HPRT1 |
7 |
31.77 |
HPRT1 |
7 |
31.49 |
|
HPRT1 |
8 |
30.76 |
HPRT1 |
8 |
30.56 |
|
HPRT1 |
9 |
32.61 |
HPRT1 |
9 |
31.84 |
|
TBP |
7 |
27.41 |
TBP |
7 |
27.23 |
|
TBP |
8 |
26.66 |
TBP |
8 |
26.62 |
|
TBP |
9 |
29.41 |
TBP |
9 |
29.02 |
- “Bar graphs of band densities normalized to ß-actin show means. Graphs show means ± SEM.”
=> “Bar graphs show means ± SEM of band densities that were normalized to ß-actin”.
These corrections have been made.
- “log2 gene expression normalized to GAPDH” => Do you mean log2 fold change or log2 relative
expression?
The graphs show log2 relative expression; this has been corrected in both figure legends.
- Figure 3: Image quality is too low
Improved images have been supplied and imported into the text with the help of the Journal editorial team.
- “RT-qPCR” => “RT-RT-qPCR”
Our experiments were RT-qPCR. This is clarified in the Methods section.
- “Bar graphs of band densities normalized to ß-actin show means. Graphs show means ± SEM.”
=> “Bar graphs show means ± SEM of band densities that were normalized to ß-actin”.
These corrections have been made.
- Lines 211-212 “To assess accelerated senescence in intestinal tissue, we used p21/waf1 as a
marker (Fig. 3A).” => as mentioned above, p21 cannot be seen as senescence marker as its
expression might be transient after radiation to induce cell cycle arrest until DNA damage is
repaired. For prolonged cell arrests, other factors such as p16 are required.
The Reviewer is correct; in some of our past studies, we had multiple time points showing sustained upregulation of p21/waf2, that better supported our hypothesis. For the minipig study, we were unable to obtain multiple time points, due to cost of the study, the specifics of the study aim, and housing availability. We have amended our conclusions based on this, as the Reviewer recommends. Additional writing is also included in the Discussion.
- Table 1: The first lines seem to be shifted, therefore, it is not possible to understand the table at
the current stage.
We have improved the alignment in the Table. We had difficulty in importing the tables into the Journal format, and we have submitted the Tables for help with importing.
- Table 2: The first lines seem to be shifted, therefore, it is not possible to understand the table at
the current stage.
We have improved the alignment in the Table. We had difficulty in importing the tables into the Journal format, and we have submitted the Tables for help with importing.
- Figure 4: Image quality is too low.
Improved images have been supplied and imported into the text with the help of the Journal editorial team.
- A: the x-axis labels are in category mode, while the plots (line-scatter) require a linear scale on
the x-axis. Please adapt the X-axis to a linear scale.
This correction has been made to the figure.
- The results for the sham-irradiated animals are missing.
We now include data from sham-irradiated, captopril-treated animals at 35 days in histological analyses, western blots, and qPCR figures.
Inclusion of the captopril sham irradiated treatment group would provide additional information regarding the effect of captopril alone on the microbiome, but this group was not included in the short time course study, as we had already completed a sham+captopril group in the long time course study. A recent publication on the effect of captopril treatment for 4 weeks had no significant effect on alpha diversity scores in the gut microbiome but did alter the microbiome composition in normal Wistar Kyoto rats (Yang et al. 2019). However, no similar studies were performed in swine or in humans. This comment is now included in the Discussion.
Discussion
- Line 356 “1.79-1.80 Gy” => “1.79-1.81 Gy”?
This has been corrected to 1.79 – 1.80 Gy.
Materials and Methods
- It seems that the study was only performed with male pigs. Please explain why.
The funding obtained for this small study included only the investigation in male animals. It is our goal to repeat our studies in female animals.
- Line 481 “1.79-1.80 Gy” => “1.79-1.81 Gy”?
This has been corrected to 1.79 – 1.80 Gy.
Table 3:
- A suitable gene identifier is missing, also the amplicon length
Unique gene identifiers are now included in the Table 3 footnote, and amplicon lengths are now included in Table 3 under the gene names within the table.
- “Sus scrofa (pig) sequences for CCL2” => “Primer sequences for the Sus scrofa (pig) genes
CCL2”
This correction has been made.
- Place the table after its appearance in the text.
This correction has been made with the help of the Journal.
- Storage temperature and during for tissue samples in RNAlater?
Tissues in RNAlater were initially stored at -20oC for 24 h, and then placed at -80oC for longer periods prior to RNA purification. This information is included in the Methods.
- What was the RIN of the RNA samples?
Our RNA quality was determined using Nano-Drop A260/A280 and A260/A230 as well as Experion analysis of the RNA. For the requested PCR revision, we have produced new RNA, and we now provide our Experion results (see Attachment 1 below). The RQI (the analysis provided by Experion instead of RIN) was typically 9.3-9.7 for our samples. We provide a sample Experion report, with the images of the RNA runs. This information is provided in the Methods.
- “RT-RT-qPCRs were performed in technical duplicates using iTaqTM Universal SYBR Green Supermix (Bio-Rad) on a CFX96 Touch Real-Time PCR Detection System (Bio-Rad) as previously described [45].” => If the cDNA was already generated in the previous step using iScript, this step describes “RT-qPCRs” and not “RT-RT-qPCRs”.
Yes, we performed RT-qPCR, not RT-RT-qPCR.
- µM => this is not a SI unit but a lab abbreviation. The correct unit is µmol/l.
This correction has been made.
- qRT-PCR => RT-qPCR
This correction has been made.
- The temperature protocol for the RT-qPCR is missing.
The temperature protocol for the RT-qPCR is now provided in the Methods section: The cycle for qPCR was: initial 95oC 2 min followed by 39 repeated cycles of 95oC 5 sec, 53oC 30 sec, 70oC 30 sec. The cycles were followed by a melt curve assay to determine purity. For all primers, an agarose gel was also run on the PCR produces to ensure that one correct size product was produced.
Figure S1
- A. What does the bar chart on the right show? The figure legend is confusing. Is it cleaved caspase?
Why is the caspase-3 active in the gut of mock-irradiated minipigs?
In Figure S1 the bar chart on the right is band density for the activated form of caspase- 3 normalized to β-actin; this has now been clarified in the figure legend. As stated in the text, the constitutive activate of caspase-3 in the GI has been previously observed, but the biological function of this activation is not known. We included this data as caspase-3 activation western blots are conspicuously missing from other reports of apoptosis of intestinal tissues; likely other researchers encountered this problem of constitutive caspase-3 activation and omitted the data to avoid controversy. We hope that by including this data, it will lead to better characterization of pig GI tissue and biology.
- B. “RT-qPCR” => “RT-RT-qPCR”
Our experiments were RT-qPCR.
Figure S3
- The meaning of the white and orange lines in the villus is unclear.
The white line indicates the measurement for the length of the villus and the orange line indicates where top 1/3 of the villus was measured. This information is now included in the figure legend.
Minor points:
Line 233 “wasn’t a significant effect” => “was no significant effect”
Line 237 “wasn’t a significant effect” => “was no significant effect”
Line 245 “wasn’t a significant effect” => “was no significant effect”
Line 245 “non- signifi-“ => “non-signifi-“
These corrections have been made.
